Manuscript prepared for Atmos. Chem. Phys.
with version 2015/04/24 7.83 Copernicus papers of the LaTeX class copernicus.cls.
Date: 25 November 2016

# Radical chemistry at a rural site (Wangdu) in the North China Plain: Observation and model calculations of OH, HO$_2$ and RO$_2$ radicals

Zhaofeng Tan[1], Hendrik Fuchs[2], Keding Lu[1], Andreas Hofzumahaus[2], Birger Bohn[2], Sebastian Broch[2], Huabin Dong[1], Sebastian Gomm[2,a], Rolf Häseler[2], Lingyan He[2], Frank Holland[2], Xin Li[2,b], Ying Liu[1], Sihua Lu[1], Franz Rohrer[2], Min Shao[1], Baolin Wang[1], Ming Wang[4], Yusheng Wu[1], Limin Zeng[1], Yinsong Zhang[1], Andreas Wahner[2], and Yuanhang Zhang[1,5]

[1]State Key Joint Laboratory of Environmental Simulation and Pollution Control, College of Environmental Sciences and Engineering, Peking University, Beijing, China
[2]Institute of Energy and Climate Research, IEK-8: Troposphere, Forschungszentrum Jülich GmbH, Jülich, Germany
[3]Key Laboratory for Urban Habitat Environmental Science and Technology, School of Environment and Energy, Peking University Shenzhen Graduate School, Shenzhen, China
[4]School of Environmental Sciences and Engineering, Nanjing University of Information Science and Technology, Nanjing, China
[5]CAS Center for Excellence in Regional Atmospheric Environment, Chinese Academy of Science, China
[a]now at: d-fine GmbH, Opernplatz 2, 60313 Frankfurt, Germany
[b]now at: College of Environmental Sciences and Engineering, Peking University, Beijing, China

*Correspondence to:* K. Lu (k.lu@pku.edu.cn), H. Fuchs (h.fuchs@fz-juelich.de)

**Abstract.** A comprehensive field campaign was carried out in summer 2014 in Wangdu located in the North China Plain. A month of continuous OH, HO$_2$ and RO$_2$ measurements was achieved. Observations of radicals by the laser induced fluorescence (LIF) technique revealed daily maximum concentrations between (5-15)$\times 10^6$ cm$^{-3}$, (3-14)$\times 10^8$ cm$^{-3}$ and (3-15)$\times 10^8$ cm$^{-3}$ for OH, HO$_2$ and RO$_2$, respectively. Measured OH reactivities (inverse OH lifetime) were 10 to 20 s$^{-1}$ during daytime. The chemical box model RACM 2 including the Leuven Isoprene Mechanism (LIM) was used to interpret the observed radical concentrations. As in previous field campaigns in China, modelled and measured OH concentrations agree for NO mixing ratios higher than 1 ppbv, but systematic discrepancies are observed in the afternoon for NO mixing ratios of less than 300 pptv (the model-measurement ratio is between 1.4 to 2 in this case). If additional OH recycling equivalent to 100 pptv NO is assumed, the model is capable of reproducing the observed OH, HO$_2$ and RO$_2$ concentrations for conditions of high VOC and low NO$_x$ concentrations. For HO$_2$, good agreement is found between modelled and observed concentrations at day and night. In case of RO$_2$, the agreement between model calculations and measurements is good in the late afternoon when NO concentrations are below 0.3 ppbv. A significant model underprediction of RO$_2$ by a factor 3 to 5 is found in the morning at NO concentrations higher than 1 ppbv, which can be explained by a

missing $RO_2$ source of $2\,ppbv\,h^{-1}$. As a consequence, the model underpredicts the photochemical net ozone production by $20\,ppbv$ per day, which is a significant portion of the daily integrated ozone production ($110\,ppbv$) derived from the measured $HO_2$ and $RO_2$. The additional $RO_2$ production from the photolysis of $ClNO_2$ and missing reactivity can explain about $10\,\%$ and $20\,\%$ of the discrepancy, respectively. The underprediction of the photochemical ozone production at high $NO_x$ found in this study is in consistent with the results from other field campaigns in urban environments, which underlines the need for better understanding of the peroxy radical chemistry for high $NO_x$ conditions.

## 1 Introduction

Air pollution in Chinese megacity regions has become an issue of great concern for citizens and the government. Ambitious restriction strategies have already been implemented for the reduction of the primary air pollutants sulfur dioxide ($SO_2$), nitrogen oxides ($NO_x$) and particular matter ($PM_{10}$) for more than a decade. Significant emission reductions of those primary air pollutants were achieved. However, high concentrations of secondary air pollutants, e.g. ozone ($O_3$) and small particles ($PM_{2.5}$) still occur and the air quality has been steadily deteriorating in some locations (Shao et al., 2006). As denoted in the Empirical Kinetics Modelling Approach (Ou et al., 2016), the reduction in primary pollutants may not directly reduce $O_3$ due to the non-linearity of atmospheric photochemistry. Thus, a critical question is to find an optimized way to control the abundance of secondary air pollutants by the reduction of primary pollutants.

As shown in a large number of studies, hydroxyl radical (OH) chemistry controls the atmospheric oxidation globally (Stone et al., 2012; Ehhalt, 1999). However, also other oxidants can be of importance on a regional scale like $NO_3$ (Brown and Stutz., 2012), Criegee intermediates (Mauldin et al., 2012) and chlorine radicals (Thornton et al., 2010). In China, studies of atmospheric oxidants are still sparse (Lu et al., 2010; Wang et al., 2015). In summer 2006, we performed two field campaigns (PRIDE-PRD2006 and CareBeijing2006) focussing on hydroxyl and hydroperoxy ($HO_2$) radical measurements in a rural area in the Pearl River Delta (PRD) and in a suburban area (Yufa) close to Beijing. The major results from these two campaigns were: 1) There were high concentrations of daytime and nighttime $HO_x$ (=OH+$HO_2$) radicals in the Chinese developed megacity regions indicating a strong atmospheric oxidation capacity. 2) The high daytime OH concentrations at high concentrations of volatile organic compounds (VOC) and low $NO_x$ concentrations could only be explained by introducing an additional OH regeneration process in the model that converts peroxy radicals to OH like NO does. An equivalent of 0.8 ppbv and 0.4 ppbv of NO was required in PRD and Beijing on average, respectively. 3) The high daytime OH concentrations at high VOC and high $NO_x$ conditions could be understood by model calculations (Hofzumahaus et al., 2009; Lu et al., 2012, 2013, 2014). A retrospective analysis (Rohrer et al., 2014) shows that the magnitude of unexplained OH concentrations observed in these two studies in China is similar to other OH observations at high VOC low $NO_x$ conditions (Tan et al., 2001; Lelieveld et al., 2008; Whalley et al., 2011).

Because isoprene was the most important OH reactant during many of these campaigns, theoretical and laboratory investigations were done to investigate its photochemical degradation. Isomerization and decomposition reactions of organic peroxy radicals formed from isoprene were found to be competitive with the reaction of peroxy radicals with NO for conditions of these campaigns (Peeters et al., 2009; Crounse et al., 2011; Fuchs et al., 2013; Peeters et al., 2014). They lead to the direct reformation of radicals and the production of hydroperoxy aldehydes (HPALD), which can photolyze and produce additional radicals. Isoprene chemistry was less important in our two field

campaigns in China 2006 compared to other campaigns that were conducted in forested areas, so that new findings in the degradation of isoprene alone could not close the gap between measured and modelled OH (Lu et al., 2012, 2013).

As a continued effort to explore the hydroxyl radical chemistry in Chinese megacity areas, OH, $HO_2$, $RO_2$ radical concentrations and OH reactivity were measured for one month at a rural site (Wangdu) in the North China Plain in summer 2014 as part of a comprehensive field campaign. Several improvements were achieved in comparison to the previous campaigns PRIDE-PRD2006 and CareBeijing2006. (1) Interference tests were performed for OH measurements applying a new

prototype chemical-modulation device. (2) Unlike before, $HO_2$ was measured without interferences by $RO_2$ species that are formed from alkenes and aromatic VOCs. (3) Total $RO_2$ was measured together with OH and $HO_2$ in contrast to the two previous campaigns. (4) In addition, the sum of $RO_2$ species that are formed from alkenes and aromatic VOCs was measured as a separate class of $RO_2$. (5) Oxygenated VOCs (e.g., formaldehyde, acetaldehyde, isoprene oxidation products) were mea-

sured, whereas such observations were missing in the previous two campaigns. All improvements provide better constraints for the interpretation of the radical chemistry. The radical measurements were obtained by a newly built, compact instrument that combines resources from Peking University and Forschungszentrum Jülich. In this paper, we report results of radical measurements and model calculations compared to results from previous campaigns investigating $HO_x$ chemistry in China.

**2  Methodology**

**2.1  Measurement site**

The campaign took place between 7 June and 8 July 2014. The measurement site is close to the town Wangdu (population 260000 inhabitants) without major industry. The Taihang Mountains are located 50 km northwest of Wangdu and the Bohai sea 200 km east. The next large city, Baoding, is

35 km northeast of Wangdu. Beijing and Shijiangzhuang, two of the largest cities in the North China Plain, are located 170 km northeast and 90 km southwest of the site, respectively. Times given in this paper are CNST (Chinese national standard time = UTC + 8 h). Sunrise was at 04:30 CNST and sunset at 20:00 CNST.

    Instruments were set-up in a botanical garden, which was surrounded by farmland. Wheat and

willows were the dominant plant species few of which were growing within 10 m of the instruments. There was no car or truck traffic in the botanical garden; the closest road was 2 km away. Most of the instruments were placed in seven sea containers. Two of them were stacked on top of three others and two more containers were placed approximately 5 m away.

## 2.2 Instrumentation

A large number of trace gases and aerosol properties were measured during this campaign. Most of the instrument inlets were placed $7\,m$ above the ground at the height of the upper containers. Table 1 summarizes the details of the trace gas measurements. OH, $HO_2$, and $RO_2$ radicals were measured by laser induced fluorescence described in detail below. The OH reactivity ($k_{OH}$), which is the inverse chemical lifetime of OH was directly measured by a laser pump and probe technique

(Lou et al., 2010; Fuchs et al., 2016b).

    Most of the inorganic trace gases ($O_3$, CO, $CO_2$, NO, and $NO_2$) were simultaneously monitored by more than one instrument. Measurements of $O_3$, CO and $CO_2$ measurements agreed well within the instrumental accuracies. $O_3$ measurements were performed by two commercial instruments using ultraviolet (UV) absorption (Environment S.A. model 41M, and Thermo Electron model 49i). Also

$SO_2$, CO, $CO_2$ concentrations were measured by commercial instruments (Thermo Electron model 43i-TLE, 48i-TLE and 410i). In addition, a cavity ring-down instrument (Picarro model G2401) provided measurements of CO, $CO_2$, $CH_4$ and $H_2O$ concentrations.

    Chemiluminescence technique was used to detect NO and also $NO_2$ after conversion to NO. Two commercial instruments were deployed by Peking University (PKU) (Thermo Electron model 42i

NO-$NO_2$-$NO_x$ analyzer), one of which (PKU-PL) was equipped with a home-built photolytic converter for the detection of $NO_2$ and the other with a catalytic converter (PKU-Mo). The $NO_2$ data from PKU-Mo were not used here, since catalytic converters can cause interferences from other nitrogen oxygen compounds (e.g., $HNO_3$, PAN). Another instrument was operated by Forschungszentrum Jülich (FZJ) (Eco Physics model TR 780, with photolytical converter for $NO_2$). Instruments

were located in the upper two containers to have inlet lengths as short as possible in order to minimize the correction for shifts in the NO to $NO_2$ ratio by the reaction of NO with $O_3$ in the inlet lines. The effect of changes of the NO to $NO_2$ ratio by peroxy radicals is negligible due to their small concentrations and their high loss rate in the inlet line. The distance between inlets was less than $5\,m$. Measurements of the two PKU instruments and the FZJ instrument differed overall by

$\pm\,20\%$, which cannot be explained by their calibration errors. The reason for this discrepancy is not clear. Calibrations of the FZJ instrument were less reproducible ($10\%$) than in previous deployments, while calibration measurements of the PKU instrument varied only by 1 to 2 %. Fortunately, the calibrations did not show a trend over time indicating that there was no accumulation of contaminations in the inlet lines. Because of the more stable calibrations of the PKU instruments, the

NO and $NO_2$ data used as model input (Section 2.4) were taken from the PKU-Mo and PKU-PL instruments, respectively. However, the difference between measurements of different instruments is considered as additional uncertainty in the $NO_2$ and NO measurements.

    Six instruments detected HONO using different techniques. Home-built instruments from FZJ (Li et al., 2014) and from PKU (Liu et al., 2016) utilized long-path absorption photometry (LOPAP). In

addition, three instruments applied cavity enhanced absorption spectroscopy (CEAS) for the detec-

tion of HONO. They were operated by the US National Oceanic and Atmospheric Administration (NOAA) (Min et al., 2016), by the Anhui Institute of Optics and Fine Mechanics (AIOFM), and by the University of Shanghai for Science and Technology (USST). A gas and aerosol collector (GAC), which is based on the wet denuder/ion chromatography technique, could also detect HONO (Dong et al., 2012). The measurements between multiple instruments agreed within 30%. HONO measurements from the FZJ-LOPAP instrument are used as model constraint, because it showed the best detection limit and temporal coverage during the campaign. Results of model calculations only change less than 10%, if either measurements by the PKU-LOPAP or NOAA-CEAS are instead used as constraint. The other CEAS HONO instruments measured only during a few days. The GAC HONO measurement is known to be affected by interferences from ambient $NO_2$ and was therefore not used here.

59 organic species were measured by a gas chromatograph (GC) equipped with a mass spectrometer and a flame ionization detector (FID) (Wang et al., 2014). This instrument provided concentrations of $C_2$ to $C_{11}$ alkanes, $C_2$ to $C_6$ alkenes, and $C_6$ to $C_{10}$ aromatics. In addition, measurements of volatile organic compounds (VOCs) were performed by a proton transfer reaction - mass spectroscopy system (PTR-MS, Ionicon). These measurements included isoprene, acetaldehyde, the sum of methyl vinyl ketone (MVK) and methacrolein (MACR), benzene, toluene, styrene, $C_8$-aromatics, $C_9$-aromatics and acetonitrile. Daytime measurements of the two instruments agreed well for those species which were detected by both instruments. During nighttime, however, PTR-MS measurements gave much larger values compared to measurements by GC for some periods and some species. The reason for that is not clear, but could have been caused by interferences by other species that occur at the same mass in the PTR-MS. Because of this uncertainty mainly measurements by GC are taken as constraints for model calculation here. Measurements of acetaldehyde, MVK and MACR were only done by PTR-MS. Formaldehyde (HCHO) was measured by a commercial instrument utilizing the Hantzsch method (Aerolaser GmbH model AL4021).

A 20 m high tower with meteorological instrumentation was set up 15 m south of the containers, where temperature, pressure, relative humidity, wind speed and wind direction were measured at two different heights (10 and 20 m). The height of the planetary boundary layer (PBL) could be estimated by a ceilometer (the minimum detectable PBL height was 200 m). Photolysis frequencies were calculated from the spectral actinic photon flux density measured by a spectroradiometer (Bohn et al., 2008), whose inlet dome was placed on top of the highest container.

More trace gases were detected, but will not be discussed in detail here: Peroxyacyl nitrates (PAN) and peroxypropionyl nitrate (PPN) were measured by gas chromatography with a electron-capture-detector (Wang et al., 2010). $H_2O_2$ was collected by a scrubbing coil collector and detected by high-performance liquid chromatography (HPLC) coupled with post-column derivatization and fluorescence detection (Hua et al., 2008). Chemical ionization mass spectroscopy (CIMS) was utilized to measure nitryl chloride ($ClNO_2$) and $N_2O_5$, but measurements were only conducted after 21 June

(Tham et al., 2016). A cavity enhanced absorption spectrometry instrument was deployed to detect glyoxal, HONO and $NO_2$ (Min et al., 2016).

Aerosol properties were characterized in detail during the campaign, but will be discussed elsewhere. Measurements included particle number density and size distribution, and also chemical composition.

## 2.3   Laser-induced fluorescence instrumentation for the detection of radicals

### 2.3.1   Instrument description

OH, $HO_2$ and $RO_2$ concentrations were measured by laser-induced fluorescence (LIF) technique. LIF is a direct method to detect OH radicals (Heard et al., 2003). In addition, $HO_2$ and $RO_2$ radicals can be detected by fluorescence after chemical conversion to OH (Fuchs et al., 2008).

    The Peking University laser-induced fluorescence instrument, PKU-LIF, was deployed in this campaign for the first time. It consists of two LIF measurement cells to detect both OH and $HO_2$. It

was built by Forschungzentrum Jülich and is therefore similar to instruments from this organization that have been described earlier (Holland et al., 1995; Hofzumahaus et al., 1996; Holland et al., 2003; Fuchs et al., 2011; Lu et al., 2012). Additionally, a third measurement cell was provided by Forschungzentrum Jülich for the detection of the sum of $RO_2$ radicals (Fuchs et al., 2008).

    The instrument consists of a laser and a measurement module (Fig. 1). The laser radiation for the

OH excitation at 308 nm is generated by a pulsed, frequency-doubled, tunable dye-laser system that is pumped by a commercial Nd:YAG laser (Spectra-Physics model Navigator) at 532 nm (repetition rate: 8.5 kHz; pulse duration FWHM 25 ns). The laser light is guided to the measurement cells, to the $k_{OH}$ instrument, and to an OH reference cell by optical fibers. The laser power is divided with a ratio of 0.6:0.32:0.08 resulting in a laser power inside the measurement cells of typically 20 mW.

The signal of the reference cell, in which a large concentration of OH is produced by pyrolysis of water vapor on a hot filament, is used as a wavelength reference and allows for the automatic correction of possible drifts of the laser wavelength.

    All components of the measurement module are housed in a weather-proof, air-conditioned box placed on top of the upper container. For the OH and $HO_2$ detection cells, ambient air is sam-

pled at a flow rate of 1 slpm (standard litre per minute, at standard conditions of 25 °C and 1 atm) through conically shaped nozzles (Beam Dynamics, orifice diameter 0.4 mm) into low pressure cells ($p = 4$ hPa). $RO_2$ is measured by a differentially pumped system consisting of a chemical conversion reactor ($p = 25$ hPa), followed by a fluorescence detection cell ($p = 4$ hPa). 7 slpm ambient air is sampled through a nozzle (orifice diameter 1.0 mm) into the reactor, half of which is sampled

through a second orifice into the fluorescence cell. Nitrogen sheath flows of 1 slpm are surrounding the gas expansions of sampled air in all fluorescence cells. Reactive gases for the conversion of

peroxy radicals can be injected via ring-shape nozzles in the fluorescence cells and via an injection needle in the $RO_2$ conversion reactor.

The laser light crosses the three fluorescence cells in a single pass. Microchannel plate photomultiplier detectors (Photek, PMT 325) are used to detect fluorescence photons collected by lens systems. The detection system is mounted perpendicular to the gas beam and laser light axis. The MCPs are gated to switch off the gain for the duration of the laser pulses. The OH fluorescence is recorded by a gated photon-counting system (Becker & Hickl, PMS 300) in a 500 ns time window starting approximately 100 ns after the laser pulse when laser stray light has dropped to an acceptable level.

The total photon count-rate is composed of the OH fluorescence, solar stray light that enters the cell through the orifice, and laser stray light. The solar stray light is detected separately during a second counting window (duration of 25 μs starting 25 μs after the laser pulse), when the OH fluorescence signal has diminished. The long integration time ensures accurate subtraction of the solar background signal, after it has been scaled to the shorter OH fluorescence counting window. The remaining other background signals are separated from the OH fluorescence by wavelength modulation of the laser. Background and fluorescence signals are measured together, when the laser wavelength is tuned on the OH absorption line, and only background signals are detected, when the laser wavelength is tuned off the absorption line. During one measurement cycle the laser wavelength is tuned to four different online and two offline positions to make sure that the maximum of the OH absorption line is captured as well as the background signal. A full wavelength cycle gives a time resolution of 32 s.

### 2.3.2 Interferences in the OH measurement

It is known that $O_3$ photolysis by 308 nm radiation with subsequent reaction of $O^1D$ with water vapor can produce artificial OH inside the measurement cell. This interference was characterized in laboratory experiments and parameterized using the laser power, and the $O_3$ and water vapor concentrations. A correction is applied that is small compared to ambient OH concentrations during daytime: 50 ppbv of $O_3$ could cause an equivalent of $3\times10^5\,cm^{-3}$ OH for typical laser power (20 mW) and water concentration (1 %) in this campaign.

Potential interferences from ozonolysis reactions and $NO_3$ have been investigated for OH and $HO_2$ detection cells that are similar to the detection cells of the PKU-LIF instrument (Fuchs et al., 2016a). No significant interference was found from the ozonolysis of simple alkenes (e.g., ethene, propene), isoprene and monoterpenes at ozonolysis reaction rates of several $ppbv\,h^{-1}$, i.e. at reaction rates that are considerably higher than found in the atmosphere. Therefore, it is not expected that measurements in this campaign are affected by ozonolysis products. Interferences from $NO_3$ were reported (Fuchs et al., 2016a). The underlying mechanism is still unknown. The magnitude of the interference is $1.1\times10^5\,cm^{-3}$ OH in the presence of 10 pptv $NO_3$. No significant interference

is expected at $NO_3$ concentrations that are predicted by model calculations for the present campaign at nighttime (average 10 pptv).

Wavelength modulation used in this work to distinguish between OH fluorescence and background signals is not capable of discriminating ambient OH signals from signals caused by artificially produced OH in the detection cells. Because interferences from unknown, internal processes have been reported for two other LIF instruments (Mao et al., 2012; Novelli et al., 2014), we have tested a chemical modulation scheme that was proposed and used by these authors. For this purpose, ambient OH is removed by an OH scavenger (propane) that is added to the sampled ambient air just before entering the fluorescence cell, so that any remaining OH signal could be attributed to internally produced OH. The propane concentration has to be chosen such that most of the ambient OH is removed while it is small enough to prevent OH losses inside the fluorescence cell. When the scavenger is replaced by nitrogen, the sum of ambient OH and possible interference OH is measured. By switching between propane and nitrogen addition, ambient OH signals can be discriminated from artifacts.

In the campaign, we applied a prototype device for chemical modulation that was temporarily attached to the OH detection cell during selected periods (Table 2). The device consisted of a Teflon tube with an inner diameter of 1.0 cm and a length of 10 cm. About 20 slpm of ambient air were drawn through the tube by a blower. 1 slpm of air was sampled into the OH detection cell. At the entrance of the Teflon tube, either propane mixed with nitrogen or pure nitrogen was injected into the air flow by a small tube (stainless steel, outer diameter 1/16"). Due to technical problems with the control electronics, the device failed to operate in the first half of the campaign. In the second part of the campaign, it showed instabilities causing an increased uncertainty in the determination of the OH scavenging efficiency.

The two signals with and without propane have contributions from ambient OH ($S_{OH}$), from the known ozone interference ($S_{O3}$) and any potential additional interference signal ($S_{int}$):

$$S_{N2} = S_{OH} + S_{O3} + S_{int} \tag{1}$$
$$S_{prop} = (1-\epsilon)S_{OH} + S_{O3} + S_{int} \tag{2}$$

$\epsilon$ is the efficiency with which ambient OH is removed when propane is added.

As long as ambient OH does not change while switching between the two measurement modes, the difference between the two signals can be used to calculate the signal from ambient OH:

$$S_{OH} = \epsilon^{-1}(S_{N2} - S_{prop}) \tag{3}$$

Together with the known ozone interference the signal that is expected to be observed in the absence of an additional interference can be calculated and compared to the total signal that is measured with no OH scavenger added ($S_{N2}$).

The knowledge of $\epsilon$ is essential for an accurate quantification of potential interferences. The removal efficiency was tested and optimized in the field using the OH calibration device as a radical

source. The value of $\epsilon$ was found to depend on the flow rates of the added gases (propane and nitrogen). Propane was added as a 5% mixture in nitrogen with a flow rate between 0.02 and 0.2 lpm (Liter per minute) which was further diluted in a carrier flow of pure nitrogen (0.04 to 0.5 lpm). The dependence of $\epsilon$ on the flow rates showed that mixing of the injected propane into the high flow of ambient air was inhomogeneous similar to results reported in Novelli et al. (2014). Because of technical difficulties with the flow regulation, the removal efficiency was re-determined before each ambient titration test. The values obtained for $\epsilon$ ranged between 80% and 97% with an accuracy of 10% (1 $\sigma$) at fixed nominal propane and nitrogen flows.

Kinetic calculations show that the added propane removes less than 0.3 % of internally produced OH. The calculation assumes that the added propane is homogeneously mixed in the sampled air, yielding an expected OH lifetime which is larger than 0.1 s and therefore much longer than the residence time (3 ms) in the low-pressure detection cell. Therefore, the propane concentrations used in the chemical-modulation tests are not expected to influence possible OH interference signals.

Another systematic error could arise from the depletion of ambient OH by wall loss in the attached Teflon tube. Calibrations of OH sensitivities with and without the chemical-modulation device only differed by 5 %, which indicates that wall loss was not important.

### 2.3.3 Measurement of $HO_2$ and possible interference

The detection of $HO_2$ is achieved by chemical conversion to OH in its reaction with NO (Hard et al., 1995). Three types of interferences are known for the current instrument design.

A small OH signal is observed when NO is injected into the fluorescence cell in the absence of ambient radicals. This background signal was regularly determined during each calibration and was stable over the entire campaign. The equivalent $HO_2$ concentration of this signal is $3 \times 10^7 \, \mathrm{cm}^{-3}$ for the NO mixing ratios applied in this campaign (see below). In addition, ambient $NO_3$ radicals can cause interferences in $HO_2$ detection similar to OH (see above). The estimated interference is $1 \times 10^7 \, \mathrm{cm}^{-3}$ at 10 pptv of $NO_3$ (Fuchs et al., 2016a), which is comparable to the detection limit.

Specific $RO_2$ radicals have the potential to be converted to OH on the same time scale as $HO_2$. Therefore, they can contribute to ambient $HO_2$ measurements (Fuchs et al., 2011). In the following, the class of interfering peroxy radicals is called $RO_2^\#$. It includes, for example, $RO_2$ species derived from alkenes, isoprene and aromatic compounds. In previous papers (e.g. Lu et al. (2012)), the quantity $[HO_2^*]$ was defined as the sum of the true $HO_2$ concentration and the interference from $RO_2$ species i, which are detected with different relative sensitivities, $\alpha_{RO_2}^i$:

$$[HO_2^*] = [HO_2] + \sum \left( \alpha_{RO_2}^i [RO_2]_i \right) \tag{4}$$

$RO_2$ radicals from alkenes, for example, have $\alpha_{RO_2}^i$ values of about 0.8, when NO is sufficiently high to achieve almost complete $HO_2$ to OH conversion in the detection cell (Fuchs et al., 2011; Lu et al., 2012). A significant reduction of the relative interference from $RO_2$ can be achieved by using

a smaller amount of added NO. Although less NO will cause a smaller $HO_2$ conversion efficiency, possible interferences from $RO_2$ will be even more strongly reduced because $RO_2$ conversion to OH requires one more reaction step with NO. For this reason, the NO concentration used for the conversion of $HO_2$ during this campaign was chosen to be significantly smaller ($\leq 20\,ppmv$) than in previous field campaigns ($500\,ppmv$) (Lu et al., 2012, 2013). At this low concentration, it is expected that interferences from $RO_2$ become almost negligible (Fuchs et al., 2011).

In order to test the importance of the remaining $RO_2^{\#}$ interference in the $HO_2$ measurements, the added NO was periodically switched between two different concentration values every few minutes. Any $RO_2^{\#}$ interference is then expected to produce a systematic difference between $HO_2$ measurements with smaller and higher NO concentrations. At the beginning of the campaign, NO mixing ratios were changed between $5\,ppmv$ and $20\,ppmv$, yielding $HO_2$ conversion efficiencies of 11% and 35%, respectively. On average, $HO_2^*$ was 15% higher when the larger NO value was applied, indicating the influence of $RO_2^{\#}$. After 14 June, the mixing ratios were switched between values of $2.5\,ppmv$ and $5\,ppmv$, giving $HO_2$ conversion efficiencies of 6% and 11%, respectively. In this case, $HO_2^*$ was on average 3% higher when the larger NO value was applied. The ratios of $HO_2^*$ measurements obtained for a pair of alternating NO concentrations showed no temporal trend or diurnal variation in each part of the campaign.

The $HO_2^*$ ratios were used to derive correction factors for the determination of interference-free $HO_2$ concentrations. For small NO concentrations as used in this work, we assume that the interference from $RO_2^{\#}$ is directly proportional to the applied NO concentration. Based on this assumption, we derived $HO_2^*/HO_2$ ratios of 1.02, 1.05, and 1.2 for the addition of 2.5, 5, and $20\,ppmv$ NO, respectively. These ratios were then used as correction factors to generate a consistent data set of interference-free $HO_2$ concentrations from the $HO_2^*$ measurements. After all, the correction was small enough that deviations from this assumption would not significantly affect our results.

### 2.3.4 Measurement of $RO_2$ and possible interference

In the $RO_2$ detection system, the chemical conversion of $RO_2$ and of $HO_2$ to OH is accomplished by a two step process as described in Fuchs et al. (2008). In the first chamber (conversion reactor), the addition of $0.7\,ppmv$ NO and $0.11\,\%$ CO at a pressure of $25\,hPa$ leads to the conversion of OH and $RO_2$ to $HO_2$. The amount of NO in the reactor is optimized for complete conversion of $CH_3O_2$ to $HO_2$. Similar conversion efficiencies apply to the majority of other atmospheric $RO_2$ species, including those resulting from OH reactions with simple alkanes, monoalkenes and isoprene (Fuchs et al., 2008). If these are the dominant $RO_2$ species, then all sampled $RO_X$ (=$OH+HO_2+RO_2$) radicals are present as $HO_2$ at the exit of the conversion reactor. In the second chamber (fluorescence cell at a pressure of $4\,hPa$), $HO_2$ is converted to OH by increasing the NO mixing ratio to $0.5\,\%$. In contrast to the pure $HO_2$ detection described above, there is no need to keep the $HO_2$ conversion efficiency small to avoid simultaneous $RO_2$ conversion. Therefore, the NO concentration is much

higher compared to the NO concentration in the $HO_2$ detection system. This measurement mode
gives the total $RO_2$ concentration when the contributions of OH and $HO_2$ measured in the other two
cells are subtracted.

The $RO_x$ system can be operated in a second mode. CO is still added to the converter causing
conversion of OH to $HO_2$, but NO is switched off, so that $RO_2$ radicals are not converted to $HO_2$.
In the fluorescence cell, however, $RO_2^\#$ species are converted to OH on the same time scale as $HO_2$
at the high NO concentration. As a result, this operational mode measures $HO_2^*$ (Eq. 4). The relative
detection sensitivities, $\alpha_{RO_2}^i$, of the $RO_x$ system in the $HO_2^*$ measurement mode were determined
in laboratory experiments for $RO_2$ radicals derived from small alkenes (e.g., ethene, propene). The
values were found to be the same as reported by Fuchs et al. (2011) for an $HO_2$ detection system
with high $HO_2$-to-OH conversion efficiency. Accordingly, other $\alpha_{RO_2}^i$ values were adopted from
Fuchs et al. (2011) and Lu et al. (2012) for these experimental conditions.

The concentration measurements of $HO_2$ (from the $HO_2$ cell) and of $HO_2^*$ (from the $RO_x$ system)
allow to estimate the total concentration of $RO_2^\#$ (Whalley et al., 2013):

$[RO_2^\#]$ = ($[HO_2^*]$ - $[HO_2]$) / $\alpha_{RO_2}^\#$

Here $\alpha_{RO_2}^\#$ denotes an average, relative detection sensitivity for $RO_2^\#$ species which contribute to
$HO_2^*$. A value of $\alpha_{RO_2}^\#$ = 0.8 ±0.2 is applied here, representing the range of specific $\alpha_{RO_2}^i$ values
for the most relevant $RO_2$ species from alkenes, isoprene, and aromatics. Any error in this average
value adds to the uncertainty of the calculated $RO_2^\#$ concentration.

Like for the $HO_2$ detection system, the presence of NO alone causes background signals of $5.0 \times$
$10^7 \, cm^{-3}$ and $3.5 \times 10^7 \, cm^{-3}$ in the operational modes with and without NO addition in the con-
version reactor. In addition, $NO_3$ causes an interference signal, which is equivalent to $1 \times 10^7 \, cm^{-3}$
$RO_2$ per 10 pptv $NO_3$ (Fuchs et al., 2016a). Measurements were corrected for the NO background
signal, but no correction was applied for potential interferences from $NO_3$, because no $NO_3$ mea-
surement was available. Model calculated $NO_3$ concentrations suggest that there was no significant
interference from $NO_3$ for conditions of this campaign.

A bias in the measurement of $RO_2$ may be caused in polluted air by peroxy radicals, which are
produced in the low-pressure converter of the $RO_2$ instrument by thermal decomposition of peroxy
nitric acid ($HO_2NO_2$), methyl peroxy nitrate ($CH_3O_2NO_2$) and PAN (Fuchs et al., 2008). In the
atmosphere, $HO_2NO_2$ and $CH_3O_2NO_2$ are in a fast thermal equilibrium with $HO_2$ and $CH_3O_2$,
respectively, together with $NO_2$. The possible interference scales with $NO_2$, which was highest dur-
ing the Wangdu campaign in the morning (median value of 15 ppbv). For this condition, according
to model calculations by Fuchs et al. (2008), $HO_2NO_2$ and $CH_3O_2NO_2$ are expected to produce
interferences of +2.6 % and +9 % for the detected $HO_2$ and $CH_3O_2$ radicals, respectively. Since
$HO_2$ and $CH_3O_2$ contributed about 50 % (measured) and 10 % (modelled) to the total $RO_x$ in the
morning, the estimated interference for measured $RO_2$ is only +2 %. The interference from PAN
decomposition in the instrument was calculated by Fuchs et al. (2008) to be 0.1 pptv per ppbv of

PAN. Since the modelled PAN concentrations for the Wangdu campaign are less than $1\,\mathrm{ppbv}$, no significant interference is expected from this compound. Another bias could be due to the perturbation of the reactor chemistry from high ambient NO concentrations (Fuchs et al., 2008). For the measurements in the $RO_x$ and $HO_2^*$ mode, the corresponding interferences are estimated to be less than +1 % and +3 %, respectively, at $15\,\mathrm{ppbv}$ NO.

### 2.3.5 Calibration and detection limits

The calibration of the LIF instrument is achieved by a radical source that provides equal concentrations of OH and $HO_2$ radicals by water vapor photolysis at $185\,\mathrm{nm}$ described in detail in Holland et al. (2003). The radical concentrations delivered by the source can be calculated from the measured water vapour concentration, the gas flow, and the intensity of the $185\,\mathrm{nm}$ radiation with a $1\,\sigma$ accuracy of 10 %. Addition of CO or $CH_4$ to the calibration gas quantitatively converts the OH into $HO_2$ or $CH_3O_2$, respectively. These modes are used for the calibration of the $HO_x$ and $RO_x$ channels, respectively (Fuchs et al., 2008).

During the campaign, calibrations were done approximately every third day. No trends with time for any of the sensitivities were observed. Thus, averaged sensitivities over the entire campaign were applied to calculate radical concentrations. The variability of the measured sensitivities is considered as an additional calibration uncertainty. The reproducibilities ($1\,\sigma$ standard deviation) of the sensitivities were 5 % for the OH cell and 5 % or 10 % for the $HO_x$ cell at high or low NO, respectively. The reproducibilities of the sensitivities of the $RO_x$ system were 7 % for the detection mode without NO in the conversion reactor and 12 % for the mode with NO.

The detection limit depends on the sensitivity, the laser power, the value of the background signal, and the integration time (Holland et al., 1995). For nighttime conditions in the absence of sunlight, the detection limits for a signal-to-noise ratio of one and a measurement time of $30\,\mathrm{s}$, a laser power of $20\,\mathrm{mW}$ during this campaign were $0.32 \times 10^6\,\mathrm{cm}^{-3}$, $0.10 \times 10^8\,\mathrm{cm}^{-3}$ and $0.11 \times 10^8\,\mathrm{cm}^{-3}$ for OH, $HO_2$ and $RO_2$, respectively. During daytime, the detection limits for OH and $HO_2$ are significantly higher, because higher background signals from solar radiation are present. The typical solar background was about $40\,\mathrm{cts/s}$ which is a factor of 20 higher than the typical background signals obtained at night. Therefore, the detection limit was reduced by a factor of 5. A shade-ring was installed during the campaign to shield the cell from direct solar radiation. The detection limit of the $RO_x$ system is not different during day- and nighttime, because no significant solar radiation can enter the fluorescence cell through the conversion reactor.

### 2.4 Model calculations

A box model is used to simulate the concentrations of OH, $HO_2$, $RO_2$ and $RO_2^{\#}$, and the total OH reactivity. The model is based on the compact Regional Atmospheric Chemical Mechanism version 2 (RACM) described in Goliff et al. (2013). This mechanisms includes 17 stable inorganic species,

4 inorganic intermediates, 55 stable organic compounds and 43 intermediate organic compounds. Compounds that are not explicitly treated in the RACM are lumped into species with similar functional groups. The assignment of organic compounds that were measured during this campaign to species in the RACM is listed in Table 3.

Some modifications were applied to the RACM. The isoprene mechanism was replaced by the more detailed mechanism listed in Table 4. It is based on the Leuven Isoprene Mechanism (LIM) proposed by (Peeters et al., 2009). Here, we use the updated LIM for bulk $RO_2$ reactions described in Peeters et al. (2014). In addition, the chemistry of the first generation products of the isoprene oxidation, MVK and MACR, and isoprene hydroperoxides (ISHP) are revised. MACR has been shown to regenerate OH via $RO_2$ isomerization and decomposition (Crounse et al., 2012; Fuchs et al., 2014). OH is also formed by the reaction of $RO_2$ from MVK with $HO_2$ with a significant yield (Praske et al., 2015). The products of the reaction of isoprene hydroperoxides formed in the reaction of isoprene $RO_2$ with $HO_2$ have been revised by Paulot et al. (2009) showing that epoxides can be formed in an OH neutral reaction. The modified RACM 2 in this work has been compared to the modified RACM-MIM-GK which was used previously for model studies of the $HO_x$ chemistry in China (Lu et al., 2012). In the present study, modelled $HO_x$ concentrations differ no more than 5% between the old and new modified RACM mechanisms. It is also noteworthy that $HO_x$ results of the modified RACM-MIM-GK agreed well with predictions of the more explicit Master Chemical Mechanism v3.2 (Lu et al., 2012).

Model calculations are constrained to measured trace gases, including inorganic species ($H_2O$, NO, $NO_2$, $O_3$, HONO, CO) and organic species (methane and non-methane organic compounds listed in Table 3). Because only the sum of methyl-vinyl-ketone (MVK) and methacrolein (MACR) were measured, a ratio of 0.6:0.4 (Galloway et al., 2011) was used to divide the sum measurement to individual species. In addition, physical parameters like photolysis frequencies, temperature and pressure are constrained to measured values.

For model calculations, the measured time series are synchronized to $5\,\text{min}$ time intervals. This is done either by averaging or by linear interpolation, if the time resolution of the measurement is shorter or longer than $5\,\text{min}$, respectively. Measurements of the two instruments for ozone and CO are combined, in order to fill data gaps.

Slightly more than 60% of the measured OH reactivity can be explained by the measured concentrations of CO, $NO_x$ and hydrocarbons during daytime. More than 90% of the OH reactivity can be explained, if also measured oxygenated VOC species are included (Fuchs et al., 2016b). Consequently, there were no large amounts of other relevant OH reactants in the atmosphere which would otherwise have contributed significantly to the measured reactivity. For this reason, long-lived product species which were not measured are constrained to zero in the model in order to avoid unrealistic build-up of additional reactivity. This constraint is consistent with the assumption that most of the measured pollutants were emitted nearby and were not photochemically aged. Only aldehydes

(ALD) are not set to zero, because they lead to the formation of reservoir species for organic per-
oxy radicals (peroxy acyl nitrates, PAN and PPN), which are kept as free parameters. In addition,
HPALD that is formed in the new isoprene chemistry is not constrained to zero. In order to avoid
unrealistic accumulation of oxygenated VOC species (mostly aldehydes), an artificial, constant loss
is added, which limits their lifetime to 24 hours.

For comparison with experimental data, the modelled concentrations of individual $RO_2$ species
are summed up in two categories which simulate the measured total $RO_2$ and $RO_2^{\#}$ concentrations
(cf. Section 2.3.4). Modelled $RO_2$ contains those species that can be detected by the measurement
system. The largest class of $RO_2$ that is not included in the calculated $RO_2$ are $NO_3$-alkene adducts
(RACM name OLND), because their reaction with NO does not produce $HO_2$. The largest concen-
tration of OLND is predicted in the early evening (approximately $1 \times 10^8 \, cm^{-3}$). In contrast, the
majority of modelled $RO_2$ during daytime consists of species which are detected. In the model, the
observable $RO_2$ species contribute with equal weight to the total $RO_2$, whereas laboratory calibra-
tions of the $RO_2$ instrument have shown slightly different (less than $\pm 20\%$) detection sensitivities
for the measured $RO_2$ species (Fuchs et al., 2008). Modelled $RO_2^{\#}$ represents a subclass of $RO_2$
species which are produced in RACM 2 from alkenes-, aromatics- and long chain (> C4) alkanes.

The relatively large uncertainty of the model calculations is a combination of uncertainties in
the measurements used as model constraints and reaction rate constants (for details see Lu et al.
(2012)). Differences in the measurements of NO (20 %) and HONO (30 %) from different instru-
ments change modelled OH concentrations by only 7 % and 10 %, if measurements from one or the
other instrument is taken as constraint. The uncertainties of measurements and modelling need to be
taken into account in the comparison. The uncertainty of radical measurement is mainly determined
by the $1\,\sigma$ measurement accuracies (OH:$\pm 11\%$, $HO_2$:$\pm 16\%$, $RO_2$:$\pm 18\%$) . A series of tests based
on Monte Carlo simulations show that the uncertainty of the model calculations is approximately
40%.

## 3 Results and Discussion

### 3.1 OH chemical modulation tests

Chemical modulation tests as described in Section 2.3.2 were conducted on 29 June (afternoon),
30 June (morning and afternoon), 2 July (afternoon) and 5 July (afternoon and evening). The time
periods of the tests and the atmospheric chemical conditions are given in Table 2. All test results
are shown in Fig. 2, where the measured OH signal $S_{N2}$ (without OH scavenger) is compared to
the sum of the expected signals from ambient OH ($S_{OH}$) and the known $O_3$ interference ($S_{O3}$).
Statistical error bars shown in Fig. 2 are derived from $1\,\sigma$ measurement precisions of $S_{N2}$ and $S_{prop}$.
In addition, the sum of $S_{OH}$ and $S_{O3}$ has a systematic error (not shown in Fig. 2), which is dominated
by the uncertainty ($\pm\,10\%$, $1\,\sigma$) of the removal efficiency ($\epsilon$) needed to calculate $S_{OH}$ (see Eq.3).

The signals $S_{\mathrm{N2}}$ (Fig. 2) are on average higher than the corresponding sum of $S_{\mathrm{OH}}$ and $S_{\mathrm{O3}}$. The differences vary within the range between $0.53\times10^6$ and $1.2\times10^6\,\mathrm{cm}^{-3}$ (Table 2) and could be the result of an unknown OH interference or of the systematic experimental error in the determination of $S_{\mathrm{OH}}$+$S_{\mathrm{O3}}$. The differences are subject not only to statistical errors, which shown as the error bars in Fig. 2 but also to the uncertainty arising from the calculation of $S_{\mathrm{OH}}$ (Eq. 3). Among all, the uncertainty in the removal efficiency ($\epsilon$) has the largest impact on the derived differences. The differences between $S_{\mathrm{N2}}$ and $S_{\mathrm{OH}}$+$S_{\mathrm{O3}}$ and their uncertainty are listed in Table 2. No correlation of differences with time of day or with the chemical conditions is observed. The differences fall quantitatively into the $2\,\sigma$ range of the accuracy of $S_{\mathrm{OH}}$+$S_{\mathrm{O3}}$ and are therefore at the limit of detection of the experimental set-up used in the campaign. Because the test results are not sufficiently accurate to draw firm conclusions about an unknown interference, the OH data in this work was not corrected for a potential interference. Instead, the differences found in Fig. 2 are treated as an additional uncertainty of the OH measurements presented in this paper.

In case of an interference, it would be a small fraction of the measured OH during daytime. The measured nighttime OH, however, would be much stronger affected. Because the existence of an unknown OH interference cannot be strictly ruled out, the interpretation of the radical chemistry will therefore concentrate on daytime conditions. More precise and accurate chemical modulation tests with an improved experimental set-up are needed in future field campaigns.

## 3.2 Meteorological and chemical conditions

Meteorological conditions were characterized by high temperatures of up to $37\,^{\circ}\mathrm{C}$ and high humidity. The wind velocity was usually below $2\,\mathrm{m/s}$. Back trajectory analysis using the NOAA HYSPLIT (Hybrid Single Particle Lagrangian Integrated Trajectory Model) model (Stein et al., 2015) showed that air masses were often transported from south or east where large city clusters are located. Solar radiation was strong during this campaign with few exceptions of hazy or cloudy days (15 to 19, 25 June and 1 to 4 July, Fig. 3).

Afternoon CO mixing ratios increased during several periods indicating accumulation of anthropogenic emissions on a regional scale. They are separated by sudden drops during rain events on 19 June and 4 July and on 27 and 28 June when clean air was transported from the north.

During the first half of the campaign, burning of agricultural waste after harvesting in surrounding fields was observed. This was confirmed by high acetonitrile mixing ratios ($>1\,\mathrm{ppbv}$) from 12 to 19 June. Biomass burning was accompanied by a reduced visibility and an increase in aerosol mass concentrations ($\mathrm{PM}_{2.5}$) with maximum values of $150\,\mathrm{\mu g\,cm}^{-3}$ on 16 June (campaign average value: $70\,\mathrm{\mu g\,cm}^{-3}$).

Time series of $O_3$ and $NO_2$ often showed trends similar to CO, but were also strongly influenced by photochemistry. Maximum daily ozone mixing ratios ranged between 100 and $140\,\mathrm{ppbv}$ depending on the strength of radiation. Because solar radiation was attenuated between 14 and 19 June

during the first pollution episode, $O_3$ peaked already on 14 June. $O_3$ was sometimes completely titrated by nitric oxide during night.

Isoprene mixing ratios exhibited a typical diurnal profile with maximum values between a few hundred $pptv$ and nearly $4\,ppbv$ in the afternoon. These values indicate that chemical conditions were also influenced by presumably local biogenic emissions.

### 3.3 Time series of measurements and model calculations

The time series of measured and modelled OH, $HO_2$, $RO_2$ and $k_{OH}$ are shown in Fig. 4. Distinct diurnal profiles are observed for all radical species. The daily maxima of OH, $HO_2$ and $RO_2$ appeared around noontime and concentrations ranged between (5-15)$\times 10^6\,cm^{-3}$, (3-14)$\times 10^8\,cm^{-3}$ and (3-15)$\times 10^8\,cm^{-3}$, respectively. On 18, 19 and 25 June and from 1 to 3 July, radical concentrations were low due to attenuated solar radiation. On 28 June, OH increased to exceptionally high concentrations of up to $3 \times 10^7\,cm^{-3}$ for a short period of time, which was accompanied by an increase of the HONO mixing ratio to $2\,ppbv$, leading to enhanced OH production from HONO photolysis. During this time, farmland next to the measurement site was treated with water and artificial nitrogen-containing fertilizer, which may have caused large, local HONO emissions.

In general, the model reproduces well the measured time series of OH, $HO_2$ and $RO_2$. Differences between modelled and measured radical concentrations are generally smaller than the combined $1\,\sigma$ uncertainties of radical measurements ($10\,\%$) and model calculations ($40\,\%$).

A closer look at the modelled and measured radical concentrations reveals some systematic trends. Modelled OH concentrations tend to be smaller than measurements during afternoon hours and modelled $RO_2$ concentrations tend to be lower in the early morning and higher in the evening than corresponding $RO_2$ measurements. In contrast, differences between modelled and measured $HO_2$ concentrations are small at all times. Because of the similarity of the model-measurement agreement for different days, further analysis of daytime radical concentrations will be done on the basis of median diurnal profiles (Section 3.4).

The OH observed at night are mostly above the limit of detection ($3\times 10^5\,cm^{-3}$) with concentrations around $5\times 10^5\,cm^{-3}$, whereas the model predicts concentrations below the limit of detection. In a few nights, the measured OH is even higher (e.g., 1-3 $\times 10^6\,cm^{-3}$ on 13 June). The reason why the measured OH values are significantly higher than the model prediction is not clear. It could be caused by missing chemistry in the model or vertical gradients in the nocturnal boundary layer as discussed in Lu et al. (2013). Furthermore, we cannot exclude an unknown interference of the same magnitude. The known interference from $NO_3$ is probably not sufficient as an explanation (Fuchs et al., 2016a), the expected interference would be $1\times 10^5\,cm^{-3}$ for this campaign, 5 times less than the averaged nighttime OH measurement. Thus, if interferences played a role, they would probably have a different origin.

The time series of measured OH reactivity shows a change on 20 June (Fig. 4). During the first two weeks, diurnal profiles of $k_{OH}$ are more structured and show higher values with maximum values of up to $40 \, s^{-1}$ compared to values after 20 June, when $k_{OH}$ is only around $10 \, s^{-1}$ in the afternoon and exhibits a less distinct diurnal profile. The first period coincides with the accumulation of pollutants like CO, nitrogen oxides and particles (Fig. 3). In addition, harvesting and biomass burning activities caused local emissions of OH reactants, which may explain the short-term increases in OH reactivity during this period, especially during nighttime, when fresh emissions are released into the shallow nocturnal boundary layer and highest OH reactivity is observed. After 20 June, biomass activities close to the measurement place were less often observed and heavy rainfall cleaned the air.

In the first period of the campaign, the model often underpredicts the measured OH reactivity, especially at night. This is likely caused by unmeasured atmospheric compounds from local emission sources like biomass burning. In the second period, the modelled and measured reactivities agree well at day and night for most of the time.

### 3.4 Median diurnal profiles

Differences between measurements and model calculations are further analyzed using median diurnal profiles with a time resolution of one hour (Fig. 5). Data are only included when measurements of all key species used as model constraints and radical measurements are available at the same time. Therefore, four days are excluded from the analysis from the entire data set. On 13 June, data gaps are larger than 6 hours for nearly all instruments. No measurements of VOCs are available on 14 June, no measurements of photolysis frequencies on 22 June and no radical measurements on 4 July. As described in Section 3.3, chemical conditions were slightly different before and after 20 June. We found similar results of model-measurement comparisons for radicals from the two periods for daytime conditions. Therefore, the following interpretation and discussion will focus on campaign averaged diurnal profiles. Chemical conditions of data included in the median profile are summarized in Table 5 and median diurnal profiles of important photochemical parameters are shown in Fig. 6.

The median diurnal profiles of the measured and modelled OH concentrations agree within their errors of $10 \%$ ($1 \, \sigma$) and $40 \%$, respectively, from sunrise to mid afternoon. When the median NO mixing ratio (cf. Fig. 6) drops gradually from 0.3 ppbv to 0.1 ppbv in the afternoon, a systematic difference evolves, with measured OH concentrations being approximately $1 \times 10^6 \, cm^{-3}$ higher than the model calculations. The discrepancy is of similar magnitude as the averaged unexplained OH determined in the chemical modulation experiments (Table 2). Thus, the overall agreement for OH would improve, if the unaccounted signal was fully considered as an OH measurement interference. However, the underestimation of OH would persist for low NO conditions if a potential unaccounted signal was subtracted. When NO concentrations are less than 100 pptv, the observed-to-modelled OH ratio would be reduced from 1.9 to 1.5, indicating that an OH source would still be missing for

low NO conditions. Although newly proposed isoprene mechanisms have the potential to enhance the OH regeneration for low $NO_x$ condition, they only have a small effect on modelled OH concentration at the conditions of this study with NO concentrations higher than $0.1$ ppbv and isoprene concentrations lower than $2$ ppbv.

In general, $HO_2$ concentrations are reproduced by the model during daytime within the com-
bined uncertainties of measurements and model calculations. Nevertheless, the model has a tendency to over-predict $HO_2$ in the afternoon. If we constrain the model to the observed $HO_2$ concentrations, the observed-to-modelled OH ratio increases from 1.6 to 1.8 for daytime averaged conditions (04:30–20:00). $RO_2$ and $RO_2^{\#}$ are significantly underestimated during the morning hours (06:00–10:00) with an observed-to-modelled ratio of 3 to 5, which is larger than the combined uncertainty
(a factor of 2). Reasons for discrepancies between measured and modelled $RO_2$ are further analyzed in Section 3.6.

Measured $k_{OH}$ is high during night, peaks in the morning ($22\,\mathrm{s}^{-1}$) and decreases to about $11\,\mathrm{s}^{-1}$ in the afternoon. Modelled $k_{OH}$ shows a relative flat diurnal profile (average over the day is $14\,\mathrm{s}^{-1}$). Whereas good agreement with measurements is achieved during daytime, measured reactivity is
higher during nighttime especially during the first part of the campaign. This is likely caused by unmeasured emitted OH reactants. A sensitivity model run, in which product species are not constrained to zero as in this model run, does not give significantly different OH reactivity in the night. A more detailed analysis of the OH reactivity in this campaign is presented in our companion paper by Fuchs et al. (2016b).

## 3.5   Correlation of OH with $j(O^1D)$

Strong correlation has been found between $j(O^1D)$ and OH radical concentrations for many field campaigns in different environments from marine to continental locations (Ehhalt and Rohrer, 2000; Brauers et al., 2001; Berresheim et al., 2003; Rohrer and Berresheim, 2006; Lu et al., 2012, 2013). A strong linear correlation is also observed for data from this campaign (Fig. 7). A linear fit between
measured OH concentrations and measured photolysis frequencies yields a slope of $4.5\times10^{11}\,\mathrm{s\,cm}^{-3}$. This value is similar to values that were derived in previous field campaigns in China in 2006 in the Pearl River Delta and Yufa (Lu et al., 2012, 2013).

The intercept of the linear fit for the campaign in Wangdu is $1.0\times10^6\,\mathrm{cm}^{-3}$, which is smaller than intercepts obtained for the data set from the campaigns in the Pearl River Delta ($2.4\times10^6\,\mathrm{cm}^{-3}$,
Lu et al. (2012)) and Yufa ($1.6\times10^6\,\mathrm{cm}^{-3}$, Lu et al. (2013)). The intercept gives an estimate of the importance of radical sources when the production of $O^1D$ from ozone is small. This includes non-photolytic sources (e.g. ozonolysis of VOCs) and photolytic processes in the early morning before $j(O^1D)$ starts to rise (Fig. 10).

Modelled OH also shows a strong dependence on $j(O^1D)$ with a slightly smaller intercept com-
pared to the fit result using the measurements.

### 3.6 Model-measurement comparison of $RO_2$

Figure 8 shows median diurnal profiles of measured $RO_2$ and $RO_2^{\#}$ together with modelled concentrations of speciated $RO_2$ radicals. The observed profiles of $RO_2$ and $RO_2^{\#}$ have similar shapes with a maximum around 14:00. In the morning hours, $RO_2$ is dominated by $RO_2^{\#}$, whereas $RO_2^{\#}$ plays only a minor role in the late afternoon and during night. The model reproduces the general behaviour of $RO_2$ and $RO_2^{\#}$ well, with very good agreement in the afternoon. However, in the morning the model underestimates $RO_2$ systematically by a significant amount of $(1\text{-}2)\times10^8\,\mathrm{cm}^{-3}$. This is mainly caused by an underestimation of $RO_2^{\#}$. After sunset in the first half of the night, the model overestimates $RO_2$. This discrepancy is apparently related to organic peroxy radicals, which do not belong to $RO_2^{\#}$.

In the group of modelled $RO_2^{\#}$ species, isoprene peroxy radicals (ISOP) make the largest contribution during daytime. Other modelled $RO_2^{\#}$ include peroxy radicals from alkenes, aromatics, long-chain ($> C_4$) hydrocarbons, and MVK and MACR. Among the $RO_2$ radicals which do not belong into the $RO_2^{\#}$ group, peroxy radicals of short-chain ($<C_5$) alkanes are dominating: methyl peroxy radicals ($MO_2$), ethyl peroxy radicals (ETHP), and peroxy radicals from $HC_3P$ (e.g., propane). Acetyl peroxy radicals ($ACO_3+RCO_3$) are also a substantial fraction of $RO_2$.

The strong underprediction of the observed $RO_2$ by more than a factor of 4 in the morning cannot be explained by the measurement errors and interferences discussed in Sections 2.3.4 and 2.3.5. In order to explore potential reasons for this underprediction, several sensitivity tests were performed. First, the impact of a faster OH to $RO_2$ conversion by an increased amount of VOC was tested (model sensitivity run S1). Second, an additional primary source of $RO_2$ was introduced into the chemical mechanism (S2). Third, the possibility of a slower removal rate of $RO_2$ was tested (S3).

The first possibility (S1) is supported by the observation that the modelled OH reactivity in the base run (S0) is smaller than the measured OH reactivity in the morning until about 09:00. If this missing reactivity is caused by unmeasured VOCs, the true $RO_2$ production from reactions of VOCs with OH would be larger than the modelled one. To fill this gap, the total concentration of the measured VOCs is increased to match the measured $k_{OH}$ in the time window from 06:00 to 09:00. The relative partitioning of the VOCs is not changed. The model run (S1) with the upscaled VOC reactivity resolves part of the $RO_2$ discrepancy until 09:00 (Fig. 5). The observed-to-modelled $RO_2$ ratio is improved from 2.8 to 1.7 without affecting the good model-measurement agreement for OH and $HO_2$. Further sensitivity tests shows that the modelled $RO_2$ is not sensitive to the speciation of the additional VOC reactivity, since the required change of $k_{OH}$ is relatively small ($< 20\,\%$). Because no missing OH reactivity is found after 09:00, the gap between measured and observed $RO_2$ cannot be explained by unmeasured VOCs later.

In sensitivity test S2, an additional primary source of $RO_2$ (OLTP) from terminal alkenes is introduced into the model. A source strength of $2\,\mathrm{ppbv\,h}^{-1}$ from 06:00 to 12:00 would be required to achieve a good model-measurement agreement (within $20\,\%$) for both $RO_2$ and $RO_2^{\#}$. The modelled

OH and $HO_2$ concentrations also increase and are slightly overpredicted by about 10 % and 20 %, respectively. This can still be considered as agreement within the error of measurements and model calculations. After 12:00 the difference between modelled and measured $RO_2$ becomes smaller than 15 %, within the range of the accuracy of $RO_2$ measurements.

A candidate for an additional primary $RO_2$ source would be reactions of VOCs with chlorine atoms, which are produced by photolysis of nitryl chloride ($ClNO_2$) (Osthoff et al., 2008). $ClNO_2$ is formed from the heterogeneous reactions of $Cl^-$ ions with nitrogen pentoxide ($N_2O_5$) and accumulates during nighttime. After sunrise, $ClNO_2$ is expected to be completely photolysed within a few hours. The resulting Cl atoms can abstract H-atoms from saturated hydrocarbons or can add to alkenes. The alkyl radicals produce $RO_2$ which in case of alkene-derived peroxy radicals carry a chlorine atom. $ClNO_2$ was measured by a CIMS instrument at the Wangdu field site from 20 June to 8 July (Tham et al., 2016). The concentrations increased during night and reached on average high values of 0.5 ppbv at 08:00, followed by a decay to zero until 11:00. In their study, Tham et al. (2016) investigated the role of $ClNO_2$ photolysis on the photochemical formation of $RO_2$ and ozone during the Wangdu campaign. They used the MCM v3.3 with an additional chlorine chemistry module by Xue et al. (2015). We repeated the study by adding the same chlorine chemistry to our modified RACM 2 mechanism and found the same additional formation rates of $RO_2$ and $O_3$ as reported by Tham et al. (2016). In our model run, a $ClNO_2$ source is assumed that leads to a linear increase of $ClNO_2$ concentrations during nighttime to a maximum value of 0.5 ppbv at 08:00 on every day. After 08:00, the modelled source is turned off. $ClNO_2$ starts to photolyze after 06:00 with a photolysis frequency that was calculated from the measured actinic flux. A maximum Cl production rate of 0.2 ppbv h$^{-1}$ is obtained at 08:00, yielding an additional $RO_2$ production with a similar rate. Compared to the additional $RO_2$ production rate required for model run S2, this is an order of magnitude too small. The mechanism is also not capable to sustain the additional $RO_2$ production during the whole morning, because $ClNO_2$ is photolytically depleted within 2-3 hours. Even if the modelled source strength is increased to match the highest $ClNO_2$ mixing ratio of 2 ppbv observed on 21 June (Tham et al., 2016), the additional primary $RO_2$ production of 0.5 ppbv h$^{-1}$ is still not sufficient. Thus, although $ClNO_2$ photolysis was a relevant radical source, it alone cannot explain the missing source of $RO_2$ radicals in the morning.

A further model test (S3) was performed, in which the rate of $RO_2$ removal was artificially reduced by decreasing the reaction rate constants between $RO_2$ and NO. Such a reduction would be justified, if the rate constant for $RO_2$+NO would be systematically too large in the model. Another reason could be a systematic measurement error of the NO concentration, or a segregation effect between $RO_2$ and NO due to inhomogeneous mixing in case of local NO emissions. In order to account for the discrepancy between modelled and measured $RO_2$ in the morning, the loss rate would have to be changed by a factor of 4, which seems unrealistically high for each of the above mentioned

possibilities. Also, there is no plausible reason why a systematically wrong rate constant or NO measurement error would appear only during morning hours.

The overprediction of $RO_2$ by the model in the evening could be related to the differences in the chemistry of $RO_2$ during day and night. Because VOC oxidation by $NO_3$ is a major contribution to $RO_2$ production at night, the inability of the model to predict $RO_2$ at night could be due to the difficulties to reproduce $NO_3$ in a box model.One complication is that the NO concentrations are close to the limit of detection of the instrument (60 pptv), which leads to a large variation in $NO_3$ concentrations in the model because of the fast reaction between $NO_3$ and NO. Assuming no $RO_2$ production from $NO_3$ chemistry would bring measured and modelled $RO_2$ into agreement.

### 3.7  NO dependence of the radical concentrations

$NO_x$ plays a crucial role in $RO_x$ chemistry due to radical propagation via peroxy radical reactions with NO and radical loss by the reaction of OH with $NO_2$ (Ehhalt, 1999). Because of these two counteracting processes, maximum OH concentrations are expected at $NO_x$ mixing ratios around 1 ppbv when other conditions controlling OH are constant.

Figure 9 shows the dependence of the measured and modelled radical concentrations on the NO mixing ratio. In order to remove the influence of the OH production strength by photolysis seen in Fig. 7, OH concentrations are normalized to $j(O^1D)$ measurements. In addition, only daytime values at NO concentrations above the detection limit of the NO instrument are included in this analysis $(j(O^1D) > 0.5 \times 10^5\,s^{-1}$, NO$> 60$ pptv). Measured OH concentrations appear to be nearly independent on the NO concentration after normalization to $j(O^1D)$. Median values of measurements are almost constant for NO mixing ratios of up to 5 ppbv. This behavior is only expected for NO mixing ratios between 0.3 and 3 ppbv as indicated by the base model calculations. Median modelled OH concentrations are nearly half of the median measured values at NO mixing ratios below 100 pptv. This discrepancy is also seen in the median diurnal profile of measured and modelled OH (Fig. 5), but it is less pronounced, because NO mixing ratios only dropped below 0.3 ppbv only for certain times and not on every day.

OH behavior similar to that shown in Fig. 9 has been reported for PRD and Yufa (Lu et al., 2012, 2013) and also for other field campaigns selected for conditions with high OH reactivity ($> 10\,s^{-1}$) (Rohrer et al., 2014). In contrast, campaigns in relatively clean air have shown a decreasing trend of OH at low NO concentrations as expected from the reduced radical recycling efficiency (Holland et al., 2003).

Measurements and model calculations show similar decreasing trends for both, $HO_2$ and $RO_2$, with increasing NO concentrations. This is expected, because the lifetime of these radical species are mainly limited by their reactions with NO. As also seen in the median diurnal profiles (Fig. 5), modelled and measured $HO_2$ concentrations agree within 20 % over the entire range of NO concentrations, whereas the measured $RO_2$ decreases less than the modelled $RO_2$ as NO increases.

At 3 ppbv NO, the modelled $RO_2$ concentrations is less than $1 \times 10^8 \, \text{cm}^{-3}$, whereas the median measured $RO_2$ is $3.5 \times 10^8 \, \text{cm}^{-3}$. As a consequence, the measured peroxy radicals yield higher calculated net ozone production rates than predicted by the model (see Section 3.8).

Two sensitivity model runs were done. In the first sensitivity run, the model did not include the updated isoprene mechanism, which is part of the base model run. The overall impact of the new 745 isoprene chemistry is rather small, the maximum increase in the median OH and $HO_2$ concentrations due to the additional OH recycling is less than $1 \times 10^6 \, \text{cm}^{-3}$ and $1 \times 10^8 \, \text{cm}^{-3}$, respectively, at NO mixing ratios lower than 0.1 ppbv. This is lower than the variability of measurements.

In the second sensitivity run, radical recycling was enhanced by introducing an artificial species X that behaves like NO, but does not produce ozone (Fig. 9). This has been successfully applied 750 to describe unexplained high OH concentration in other campaigns (Rohrer et al., 2014) including our previous observations in China (Hofzumahaus et al., 2009; Lu et al., 2012, 2013). Similar to the observations in the previous campaigns, a constant mixing ratio of X would bring modelled and measured OH into agreement for the entire range of NO concentrations (Fig. 9). Here, the concentration of X needs to be equivalent to 100 pptv NO. Modelled $HO_2$ and $RO_2$ concentrations 755 do not change much, if this mechanism is applied.

OH concentrations in this campaign are better predicted by the base model compared to our previous field campaigns that were conducted in China. In all three campaigns, median diurnal profiles of measured and modelled OH agree in the morning, but measured median OH starts to be increasingly higher than modelled OH after noontime. In this campaign, the difference is a factor of 1.4 at 16:00 760 and a factor of two at sunset (20:00). Differences are within the $2\sigma$ uncertainty of measurements for most of the time. In contrast, the difference was a factor of 2.6 to 4.5 in previous campaigns for higher OH reactivity conditions. Consequently, also the amount of additional recycling that is required to bring modelled and measured OH into agreement is less in this campaign (100 pptv NO equivalent) compared to Yufa (400 pptv) and PRD (800 pptv) in 2006. The major differences be- 765 tween this campaign and the others are: (1) OH concentrations in this campaign are smaller. (2) NO mixing ratios (100 pptv) were lower in previous campaigns, reducing the OH recycling efficiency from the reaction of peroxy radicals with NO. (3) Measured OH reactivity is around $12 \, \text{s}^{-1}$ in this campaign, but was at least 50 % larger in the other campaigns.

### 3.8 Ozone production rate

Peroxy radical measurements allow the calculation of net ozone production (Mihelcic et al., 2003). The photolysis of $NO_2$ produces $O_3$ and NO. Because $O_3$ can also be consumed in the back-reaction of NO to $NO_2$, net ozone production is only achieved, if the reformation of $NO_2$ does not involve $O_3$. This is the case, if peroxy radicals ($HO_2$ and $RO_2$) react with NO. Therefore, net ozone production can be calculated from the reaction rate of peroxy radicals with NO using measured and modelled 775 peroxy radical concentrations (Fig. 5). Production ($P(O_3)_{net}$) is reduced by the loss of $NO_2$ via its

reaction with OH and further losses of ozone ($L(O_3)$) by photolysis and reactions with OH, $HO_2$ and alkenes:

$$
\begin{aligned}
P(O_3)_{net} &= k_{HO_2+NO}[HO_2][NO] + \sum (k^i_{RO_2+NO}[RO_2^i][NO]) \\
&\quad - k_{OH+NO_2}[OH][NO_2] - L(O_3)
\end{aligned}
\tag{5}
$$

$$
\begin{aligned}
\text{780} \qquad L(O_3) &= \left( \theta\, j(O^1D) + k_{OH+O_3}[OH] + k_{HO_2+O_3}[HO_2] \right)[O_3] \\
&\quad + \left( \sum (k^i_{alkene+O_3}[alkene^i]) \right)[O_3]
\end{aligned}
\tag{6}
$$

$\theta$ is the fraction of $O^1D$ from ozone photolysis that reacts with water vapor.

The calculation of the net ozone production from the measured concentration of total $RO_2$ is complicated by differences in the reaction rate constants of NO with different $RO_2$ species. An ef-
fective rate constant is determined from the rate constants of the different $RO_2$ species in RACM 2 weighted by their relative abundance calculated by the model for each instant of time. The effective rate constant increases in the morning and reaches a maximum $8.5 \times 10^{-12}\,\mathrm{cm^3\,s^{-1}}$ in the afternoon and decreases to a value of $6.5 \times 10^{-12}\,\mathrm{cm^3\,s^{-1}}$ after dusk. For comparison, the rate constant for the reaction of $CH_3O_2$ with NO is $7.5 \times 10^{-12}\,\mathrm{cm^3\,s^{-1}}$. A systematic underestimation of the calculated
ozone production rate may arise from $RO_2$ species, which react with NO and form $NO_2$, but do no produce $HO_2$. Such $RO_2$ species would possibly contribute to the ozone formation, but are not detected in our instrument. As explained in Section 2.4, this behavior is found in the Wangdu campaign for peroxy radicals which are formed by reactions of alkenes with $NO_3$. However, because $NO_3$ is easily photolyzed, these particular peroxy radicals do not play a role during daytime and do
not contribute to photochemical ozone production.

Net ozone production has a distinct diurnal profile that peaks in the morning (Fig. 5). The peak value of $19\,\mathrm{ppbv\,h^{-1}}$ (median) derived from measurements is higher than that calculated in the model ($14\,\mathrm{ppbv\,h^{-1}}$) and shifted to earlier times (Fig. 5). The variability of this peak value is much larger than seen in the model with values up to several ten $\mathrm{ppbv\,h^{-1}}$.

If the diurnal ozone production rates are integrated for daytime (04:30 - 20:00), the model yields about $20\,\mathrm{ppbv}\,O_3$ less than the experimental value of $110\,\mathrm{ppbv}$ derived from the radical measurements. The difference between observed and modelled ozone production is mainly caused by the underestimation of the modelled $RO_2$ concentration in the morning. As discussed in Section 3.6, two generic mechanisms may partly explain the discrepancy. One possibility are unmeasured VOCs,
which would explain the model underestimation of the OH reactivity in the morning and would increase $RO_2$ by their reactions with OH. Model run S1 with adjusted VOCs shows a slightly improved agreement of the modelled and measured ozone production rates (Fig. 5), but enhances the daily integrated ozone production only by $4\,\mathrm{ppbv}$. The other possibility is an additional primary $RO_2$ source of $2\,\mathrm{ppbv\,h^{-1}}$, which is considered in model run S2. It would enhance the daily integrated
ozone production by $30\,\mathrm{ppbv}$, which is on the order of magnitude of the $P(O_3)$ underestimation.

As also mentioned in Section 3.6, one possibility for an additional primary $RO_2$ source is the reaction of VOCs with chlorine atoms from $ClNO_2$ photolysis, which is not considered in RACM 2. With a maximum $ClNO_2$ concentration of 0.5 ppbv in the morning, an additional daily integrated ozone production of about 2 ppbv is calculated. It should be noted that $RO_2$ radical species, which are produced by additional reactions of chlorine atoms with alkenes, may behave kinetically different than $RO_2$ radicals from OH reactions. In the chlorine chemistry module that we adopted from Xue et al. (2015), Cl-substituted $RO_2$ radicals have the same rate constants like OH-substituted $RO_2$ radicals, because kinetic data are missing for Cl-substituted compounds. It is, however, unlikely that this simplification has a strong influence on the calculated net ozone production.

During the first period of the campaign (8 June to 14 June) daily maximum ozone mixing ratios increased from 50 to 150 ppbv (Fig. 3). However, the connection between the photochemical ozone production rate and ozone concentrations measured over several days at a distinct location is complicated. Additional ozone loss processes need to be taken into account, for example deposition and indirect loss via reactive nitrogen chemistry during the night ($NO_3$ and $N_2O_5$) that are not included in Eq. 5 and 6. Furthermore, the effect of high ozone production in the morning on midday ozone mixing ratios is reduced due to the dilution by the increase of the boundary layer height. Also regional transportation of ozone can be of importance, if the spatial distribution of ozone production and/or loss processes is inhomogeneous. The cumulative ozone production observed during the first period of the campaign is approximately 700 ppbv. This high total ozone production indicates that most of the locally produced ozone was removed by transport or deposition.

Other $HO_x$ field studies have also found that models underpredicted the observed ozone production rate in urban atmospheres (Martinez et al., 2003; Ren et al., 2003; Kanaya et al., 2008; Mao et al., 2010; Kanaya et al., 2012; Ren et al., 2013; Brune et al., 2016; Griffith et al., 2016). In these studies, the observed production rates were determined from measured $HO_2$ concentrations only, without the contribution of $RO_2$ for which measurements were not available. In general, the ozone production from $HO_2$ was underpredicted by chemical models at NO mixing ratios greater than 1 ppbv, reaching a factor of about 10 between 10 ppbv and 100 ppbv NO. In campaigns before 2011, unrecognized interferences from $RO_2^{\#}$ species may have contributed to the deviation between measurement and model results. The interference, however, is expected to account for less than a factor of 2, because $HO_2$ and $RO_2$ concentrations are approximately equal (Cantrell et al., 2003; Mihelcic et al., 2003) and $RO_2^{\#}$ is only a fraction of the total $RO_2$ (e.g., Fig. 5). This expectation has been confirmed in recent studies, where the interference was taken into account and the significant underprediction of the ozone production from $HO_2$ still persists (Ren et al., 2013; Brune et al., 2016; Griffith et al., 2016). During the CalNex-LA 2010 campaign in Pasadena (California), part of the discrepancy could be explained by unmeasured VOCs, which were recognized as missing OH reactivity (Griffith et al., 2016). Another major reason for the $HO_2$ underprediction could be an incomplete understanding of the $HO_2$ chemistry at high $NO_x$ concentrations (Ren et al., 2013;

Brune et al., 2016; Griffith et al., 2016). Similar arguments as for the underprediction of $HO_2$ apply to $RO_2$. Whalley et al. (2016) have pointed out that modelled $RO_2$ and the associated ozone production could be severely underestimated (60 %) in the London atmosphere due to the presence of larger VOCs (mainly monoterpenes). In the Wangdu campaign, missing reactivity from unmeasured VOCs is much smaller. As shown above, unmeasured VOCs caused an underprediction of the daily ozone production of less than 5 %.

Total photochemical ozone production rates were directly measured in a sunlit environmental chamber during the SHARP campaign in Houston (Texas) 2009 (Cazorla et al., 2012; Ren et al., 2013). The comparison with ozone production rates determined from measured $HO_2$ and from modelled $HO_2$ and $RO_2$ suggests that the model underestimated both $HO_2$ and $RO_2$ at high $NO_x$ in the morning. The underprediction of the daily ozone production was a factor of 1.4. At Wangdu, we find an underprediction of the daily ozone production by a factor of 1.2, which is mainly caused by an underprediction of $RO_2$. In conclusion, all field studies indicate that the photochemical formation of ozone in a polluted urban atmosphere is not well understood either due to incomplete chemical characterization of the air composition, or incomplete understanding of the peroxy radical chemistry at high $NO_x$.

### 3.9 Budget analysis based on model results

The budget analysis for OH, $HO_2$ and $RO_2$ radicals is based on the results of model calculations. There are two classes of radical reactions. On the one hand, $RO_x$ radicals are produced or destroyed by reactions, in which $RO_x$ radical species are not reactants and products at the same time. On the other hand, $RO_x$ species are converted into each other by radical recycling reactions. In polluted air during daytime, the conversion reactions are fast, so that the $RO_X$ species are in an equilibrium. Under these conditions, the impact of primary production and destruction is similar on all radical species. The partitioning of $RO_X$, however, depends on the relative rates of the conversion reactions.

### 3.9.1 Primary radical production and destruction

Median diurnal profiles of primary radical production and destruction rates of $RO_x$ radicals are shown in Fig. 10. Highest turnover rates occur after noontime reaching maximum values around $5\,\mathrm{ppbv\,h^{-1}}$. HONO, $O_3$ and HCHO photolysis account for approximately two third of the daytime radical production. HONO, $O_3$ and HCHO concentrations as well as their photolysis frequencies are well constrained by measurements. HONO photolysis alone is the most important single primary source with maximum values of nearly $2\,\mathrm{ppbv\,h^{-1}}$ at 13:00, 38 % of the total radical production. $O_3$ photolysis contributes 15 % to the total radical production rate. Formaldehyde photolysis is a major source for $HO_2$ accounting for 18 % of total daytime primary production.

Other production processes of OH includes alkene ozonolysis, which also produces $HO_2$ and $RO_2$. The remaining part of the daytime production can be attributed to the photolysis of carbonyl compounds.

Recent findings in the understanding of the oxidation of isoprene found that photo-labile hydroxperoxy aldehydes (HPALD) can be formed in environments where radical recycling via NO is not efficient (Peeters et al., 2014). HPALD photolysis can be a significant radical source in this case. In this campaign, this reaction is almost negligible (Fig. 10), because modelled HPALD concentrations are only around $100\,pptv$.

In the morning (till 10:00), the major loss of $RO_x$ is the reaction of OH with $NO_2$. At later times, radical destruction is dominated by the loss via peroxy radical self-reactions, $HO_2+HO_2$, $HO_2+RO_2$ and $RO_2+RO_2$. $HO_2$ and $RO_2$ concentration values and their diurnal profiles are similar. Because the reactions of $HO_2$ with $RO_2$ have the largest reaction rate constant of the three types of peroxy radical self-reactions, these reactions make the largest contribution. The effect of radical destruction by $RO_2$ self-reactions could be underestimated in the model, because only reactions of $RO_2$ with methyl-peroxy radicals and acetyl peroxy radicals are included in the RACM mechanism.

The reaction of OH with NO is the only known gas phase production of HONO, which can compensate the OH production by HONO photolysis. During this campaign, however, HONO formation in the gas phase is always much smaller compared to HONO photolysis making HONO a net source of OH. This also means that the high HONO concentrations during the day cannot be explained by production from the reaction of OH with NO. The importance of HONO photolysis to $HO_x$ chemistry has been reported from urban to forest environments (Dusanter et al., 2009; Mao et al., 2010; Griffith et al., 2013; Kim et al., 2014). The observation of an unusual high HONO concentration of $2\,ppbv$ at noon of 28 June (Fig. 3), when the nearby agricultural field was treated with artificial nitrogen fertilizer, suggests that HONO emissions from surrounding farmland may have played an important role at the measurement site in Wangdu. An imbalance of the two gas-phase reactions of HONO has also been found in many other field campaigns, for example in previous field campaigns in China in 2006 (Li et al., 2012). Heterogeneous formation of HONO is thought to explain part of the missing daytime source (VandenBoer et al., 2014, and ref. therein) and photolysis of particulate nitrate is proposed to be of potential importance for tropospheric HONO production (Ye et al., 2016).

Further radical terminating OH losses include reactions with unsaturated dicarbonyls (DCB1, DCB2, DCB3) and acetyl nitrate species (PAN, MPAN, etc) in RACM 2.

Compared to our previous campaign in Yufa in 2006, the primary radical production in this campaign is significantly less in the morning mainly because of smaller OH production from HONO photolysis. In the afternoon, however, radical production was mainly due to ozone and formaldehyde photolysis in Yufa. The relative contributions of radical destruction processes are similar in this cam-

paign compared to Yufa, but radical loss due to reactions with nitrogen oxides is less important in the morning and slightly enhanced in the afternoon in this campaign.

### 3.9.2 Radical propagation reactions

Figure 11 shows the distribution of turnover rates of radical recycling reactions. These conversion reactions establish the partitioning of total $RO_x$ species into OH, $HO_2$ and $RO_2$.

The conversion of OH to $HO_2$ (43 % of the total OH destruction rate) is dominated by the reaction of OH with CO and HCHO contributing 25 % and 13 % to the total OH destruction during daytime. Isoprene and its oxidation products (MVK and MACR) are the dominant organic OH reactants in 925 the afternoon. In contrast, alkenes and aldehydes reactions with OH dominate the conversion from OH to $RO_2$ in the morning.

The radical recycling from $RO_2$ to $HO_2$ and also from $HO_2$ to OH is mainly driven by NO reactions. NO reactions with methyl peroxy radicals ($MO_2$) and isoprene derived radicals (ISOP) each account for 26 % of the total conversion rate of $RO_2$ to $HO_2$ during daytime. Alkane (ALKAP) 930 and alkene (ALKEP) derived peroxy radicals contribute another 20 % and 13 %, respectively. Their relative importance is largest in the morning.

Acyl peroxy radicals ($ACO_3$ and $RCO_3$) do not directly convert to $HO_2$, but form other $RO_2$ species ($MO_2$ and ETHP in RACM). A second reaction step with NO is required to form $HO_2$. Therefore, they are not included in the budget in Fig. 11. However, this conversion reaction con- 935 tributes to ozone production as discussed above. The daytime average turnover rate of this type of conversion reaction is $0.9\,\mathrm{ppbv\,h^{-1}}$.

Direct conversion of $RO_2$ radicals to $HO_2$ and OH by isomerization reactions with subsequent decomposition has been found to be competitive with radical recycling via reactions with NO in the isoprene oxidation mechanism (Peeters et al., 2014; Crounse et al., 2012). The effective isomer- 940 ization rate of isoprene derived $RO_2$ is $0.01\,\mathrm{s^{-1}}$ for conditions of this campaign in the afternoon hours (temperature: 303 K). This loss rate is small compared to the loss of isoprene derived $RO_2$ via the reaction with NO. The average NO mixing ratio is $0.19\,\mathrm{ppbv}$ (for the subset of days shown in Fig. 11) giving a loss rate of $0.04\,\mathrm{s^{-1}}$. Therefore, only 20 % of $RO_2$ from isoprene (ISOP) un- dergoes isomerization, so that radical recycling from ISOP to $HO_2$ via isomerization is small. This 945 also explains, why HPALD photolysis as primary $RO_x$ source is not important in this campaign. In contrast to $RO_2$ from isoprene, one $RO_2$ species from MACR (MACP) nearly exclusively isomer- izes for afternoon conditions of the campaign. However, the overall impact of this radical recycling reaction is also small, because the median production rate of MACP is only $0.14\,\mathrm{ppbv\,h^{-1}}$ in the afternoon.

The maximum turnover rate of recycling reactions is slightly shifted to earlier times compared to the maximum turnover rate of primary radical production. This is mainly due to the dominance of conversion reactions of $RO_2$ and $HO_2$ with NO. This can be best seen in the median diurnal

profile of the $HO_2$ conversion to OH, which peaks earlier than the OH conversion to $HO_2$ and $RO_2$ (Fig. 11, lower panel). Because the total OH production and destruction rates equal in the model calculation, this imbalance is compensated by the larger primary OH production (Fig. 10).

Compared to the turnover rates in Yufa 2006, radical conversion is less strong in the morning in this campaign, mainly due to smaller peak NO concentrations leading to a reduced reformation of OH from $HO_2$. This is accompanied by lower $HO_2$ production in the reaction of OH with formaldehyde. In the afternoon, the strength of radical conversion reaction is similar in both campaigns.

## 4  Summary and conclusions

A comprehensive set of measurements was achieved to characterize the photochemistry at the rural site Wangdu in the North China Plain in 2014. Air pollution was likely transported from surrounding industrial areas and farmland in the North China Plain and few days were influenced by clean air coming from the North.

A new LIF instrument was used to measure concentrations of OH, $HO_2$ and $RO_2$, and a special group of organic peroxy radicals ($RO_2^{\#}$) which are produced from alkenes and aromatics. Furthermore, total OH reactivity was measured by a laser pump-and-probe instrument. In order to test, if OH measurements included artifacts from OH production inside the measurement cell, chemical modulation tests were performed. These tests identified unexplained OH signals equivalent to $(0.5\text{-}1) \times 10^6 \, \mathrm{cm}^{-3}$ with a systematic experimental $1\,\sigma$ uncertainty of $0.5 \times 10^6 \, \mathrm{cm}^{-3}$. Given this uncertainty, the unexplained OH signal may have been caused by an experimental bias of the chemical modulation setup, but also an unknown OH interference cannot be excluded. In case of an interference, its contribution to the maximum OH concentration would have been only 10%, so that it would have a minor impact on the interpretation of daytime OH measurements. However, it cannot be excluded that nighttime OH measurements were significantly affected by interferences. An improved setup of this system will be used in future field campaigns.

Daily maximum concentrations of OH, $HO_2$ and $RO_2$ ranged from $5 \times 10^6$ to $15 \times 10^6 \, \mathrm{cm}^{-3}$, $3 \times 10^8$ to $14 \times 10^8 \, \mathrm{cm}^{-3}$ and $3 \times 10^8$ to $15 \times 10^8 \, \mathrm{cm}^{-3}$, respectively. Model calculations using a modified RACM 2 mechanism reproduce the measured radical concentrations generally well in this campaign. The modified RACM 2 contains an extension based on recent findings in the isoprene chemistry (Peeters et al., 2014; Crounse et al., 2012), which leads to a small increase of the modelled OH for the conditions of this campaign.

The model-measurement comparison for OH shows a tendency to less good agreement at low NO concentrations. At concentrations above $0.3\,\mathrm{ppbv}$ NO, OH is well described by the model, but is increasingly underpredicted at lower NO in the afternoon by up to a factor of 2. The unexplained OH signals from the chemical modulation test cannot explain this trend. Introduction of an additional radical recycling process which has the same effect as $100\,\mathrm{pptv}$ NO can close the gap between

modelled and measured OH, but the nature of the process remains unknown. This behaviour is qualitatively in agreement with previous results from two field campaigns in China, in the Pearl River Delta and in the North China Plain, where the required equivalent NO is 800 pptv and 400 pptv (Lu et al., 2012, 2013).

An opposite trend is found for $RO_2$ radicals. At higher NO concentrations in the morning, the model shows an underprediction of the measured $RO_2$, which reaches a factor of 10 at about 4 ppbv NO. The underprediction is mainly related to $RO_2^{\#}$ species, whose concentrations were half of the total $RO_2$ concentrations. The reaction of OH with unknown VOCs, estimated from missing OH reactivity, can explain part of the $RO_2$ discrepancy until 9:00, but not later in the morning. Good agreement between measured and modelled $RO_2$ and $RO_2^{\#}$ can be achieved by assuming an additional primary source of 2 ppbv $h^{-1}$ of $RO_2$ (from alkenes) until noon. Reactions of VOCs with chlorine atoms from the photolysis of $ClNO_2$ were a likely source of additional $RO_2$ after sunrise, but the measured $ClNO_2$ concentrations ($< 2$ ppbv) reported by Tham et al. (2016) can explain only (10-20)% of the required additional $RO_2$ source early in the morning. Another source which sustains additional $RO_2$ production until noon is therefore needed.

As a consequence of the model underprediction of $RO_2$, the total net ozone production from $HO_2$ and $RO_2$ radicals is also underestimated by the model. The median measured concentrations of $HO_2$ and $RO_2$ yield a daily integrated ozone production of 110 ppbv, which is 20 ppbv more than predicted by the modified RACM 2. About 10 % of the discrepancy can be explained by $ClNO_2$ chemistry during the Wangdu campaign. The underprediction of the photochemical ozone production at high $NO_x$ in the morning is in general agreement with other studies in urban environments, underlining the need for better understanding of the peroxy radical chemistry in polluted air.

Radicals are primarily produced by photolysis reactions and radical loss is dominated by reactions with nitrogen oxides in the morning and peroxy radical self-reactions in the afternoon. This is similar to our previous campaign 2006 in Yufa that is also located in the North China Plain (Lu et al., 2013). OH production from HONO photolysis in the afternoon was the largest primary radical source in this campaign. Because NO concentrations are lower than in 2006 in the morning, radical conversion rates are smaller. Higher OH concentrations and OH reactivity measured in 2006 and smaller OH recycling from the reaction of $HO_2$ with NO in the afternoon led to the need of a larger enhancement of the radical recycling efficiency for the campaign in 2006 compared to results from this campaign.

*Acknowledgements.* We thank the science teams of Wangdu-2014 Campaign. This work was supported by the Strategic Priority Research Program of the Chinese Academy of Sciences (grant no. XDB05010500), the National Natural Science Foundation of China (Major Program: 21190052 and Innovative Research Group: 41121004), the Collaborative Innovation Center for Regional Environmental Quality, the EU-project AMIS (Fate and Impact of Atmospheric Pollutants, PIRSES-GA-2011-295132). The authors gratefully acknowledge

the NOAA Air Resources Laboratory (ARL) for the provision of the HYSPLIT transport and dispersion model and/or READY website (http://www.ready.noaa.gov) used in this publication.

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

**Table 1.** Measured quantities used for data analysis and model calculations.

| Parameters | Measurement technique | Time resolution | Detection limit [a] | $1\sigma$ Accuracy |
|---|---|---|---|---|
| OH | LIF [b] | 32 s | $0.32 \times 10^6\ \text{cm}^{-3}$ | $\pm 11\%$ |
| $HO_2$ | LIF [b c] | 32 s | $0.10 \times 10^8\ \text{cm}^{-3}$ | $\pm 16\%$ |
| $RO_2$ | LIF [b c] | 32 s | $0.05 \times 10^8\ \text{cm}^{-3}$ | $\pm 18\%$ |
| $k_{OH}$ | LP-LIF [d] | 180 s | $0.3\ \text{s}^{-1}$ | $\pm 10\% \pm 0.7\ \text{s}^{-1}$ |
| photolysis frequency | spectroradiometer | 20 s | [e] | $\pm 10\%$ |
| $O_3$ [f] | UV photometry | 60 s | 0.5 ppbv | $\pm 5\%$ |
| NO [g] | chemiluminescence | 180 s | 60 pptv | $\pm 20\%$ |
| $NO_2$ [g] | chemiluminescence[h] | 60s | 300 pptv | $\pm 20\%$ |
| HONO [i] | LOPAP[j], CEAS[k] | 30s | 7 pptv | $\pm 20\%$ |
| CO, $CH_4$, $CO_2$, $H_2O$ | cavity ring down | 60 s | [l] | [m] |
| $SO_2$ | pulsed UV fluorescence | 60 s | 0.1 ppbv | $\pm 5\%$ |
| HCHO | Hantzsch fluorimetry | 60 s | 25 pptv | $\pm 5\%$ |
| volatile organic compounds[n] | GC-FID/MS [o] | 1 h | 20 to 300 pptv | $\pm 15$ to $20\%$ |
| volatile organic compounds[p] | PTR-MS | 20 s | 0.2 ppbv | $\pm 15\%$ |

[a] signal to noise ratio = 1

[b] laser induced fluorescence

[c] chemical conversion via NO reaction before detection

[d] laser photolysis - laser induced fluorescence

[e] process specific, 5 order of magnitudes lower than maximum in noon time

[f] $O_3$ was measured by two photometers (Environment S.A. (41M) and Thermo (49i)); data were taken from the Thermo (49i) instrument, which agreed well with the data from the Environment S.A. instrument (see text).

[g] NO and $NO_2$ were measured by three chemiluminescence instruments (Eco Physics CLD TR780, and two Thermo (42i-TL)); data were taken from the Thermo (42i-TL) instruments which agreed well with each other; the data accuracy represents the unexplained difference between the data from the Thermo and Eco Physics instruments (see text).

[h] photolytical conversion to NO before detection, home built converter

[i] HONO was measured by two different, home-built (FZJ, PKU) LOPAP instruments and one CEAS instrument (NOAA); data were taken from the FZJ-LOPAP instrument; the data accuracy represents the unexplained differences between the data of the three instruments (see text).

[j] long-path absorption photometry

[k] cavity enhanced absorption spectrometer

[l] species specific, for CO: 1 ppbv; $CH_4$:1 ppbv; $CO_2$: 25 ppbv; $H_2O$: 0.1 % (absolute water vapor content)

[m] species specific, for CO: 1 ppbv; $CH_4$:$\pm 1$ ppbv; $CO_2$: $\pm 25$ ppbv; $H_2O$: $\pm 5\%$

[n] VOCs including $C_2$-$C_{11}$ alkanes, $C_2$-$C_6$ alkenes, $C_6$-$C_{10}$ aromatics

[o] gas chromatography equipped with mass spectrometer and a flame ionization detector

[p] OVOCs including acetaldehyde, methyl-vinyl ketone and methacrolein

**Table 2.** Unexplained OH signal and chemical conditions during the OH interference tests. The mean value and the $1\sigma$ standard deviation of the unexplained OH signal are calculated from the differences between $S_{N2}$ and $S_{OH}+S_{O3}$ shown in Fig. 2 for each test. The differences are expressed as equivalent ambient OH concentrations (see text).

| # | Date | Time / CNST | OH /$10^6$cm$^{-3}$ | $k_{OH}$/s$^{-1}$ | NO/ppbv | ISO/ppbv | O$_3$/ppbv | T/$^{\circ}$C | Unexplained Signal/$10^6$ cm$^{-3}$ |
|---|------|-------------|---------------------|-------------------|---------|----------|------------|---------------|-------------------------------------|
| 1 | 06.29 | 13:00-15:00 | 7.0 | 15.2 | 0.16±0.11 | 2.8 | 126 | 34 | 0.65(±0.34) |
| 2 | 06.30 | 09:50-11:00 | 10.4 | 15.4 | 1.39±0.51 | 2.1 | 81 | 31 | 0.97(±0.14) |
| 3 | 06.30 | 14:40-16:10 | 8.5 | 8.8 | 0.14±0.05 | 2.0 | 110 | 34 | 1.15(±0.21) |
| 4 | 07.02 | 10:50-11:30 | 4.6 | 10.0 | 1.19±0.27 | n/a[a] | 52 | 26 | 0.74(±0.24) |
| 5 | 07.05 | 16:30-17:40 | 3.3 | 9.2 | 0.08±0.02 | 1.6 | 94 | 32 | 0.99(±0.04) |
| 6 | 07.05 | 18:00-21:00 | 1.5 | 16.7 | 0.02±0.03 | 1.4 | 77 | 31 | 0.53(±0.30) |

[a] No VOC measurement during the chemical modulation experiment

**Table 3.** Assignment of measured VOCs to species in the RACM 2 (Goliff et al., 2013).

| RACM | Measured hydrocarbons |
|---|---|
| CH4 | methane |
| ETH | ethane |
| HC3 | propane, i-butane, n-butane, 2,2-dimethybutane |
| HC5 | i-pentane, n-pentane, cyclopentane, n-hexane, 2,3-dimethylbutane, 2-methylpentane, 3-methylpentane, n-heptane, 2,4-dimethylpentane, 2,3-dimethypentane, methylcyclopentane, 2-methylhexane, MTBE |
| HC8 | cyclohexane, 3-methylhexane,2,2,4-trimethylpentane, 2,3,4-trimethylpentane, n-heptane, methylcyclohexane, 2-methylheptane, 3-methylheptane, n-octane, n-nonane, n-decane |
| ETE | ethene |
| DIEN | 1,3-butadiene |
| OLI | trans-2-butene, cis-butene, trans-2-pentene, cis-2-pentene |
| OLT | propene,1-butene, i-butene, 1-pentene, 1-hexene, styrene |
| ACE | ethyne |
| ISO | isoprene |
| BEN | benzene |
| TOL | toluene, ethylbenzene, i-propylbenzene, n-propylbenzene |
| XYM | m-ethyltoluene, 1,3,5-trimethylbenzene, 1,2,4-trimethylbenzene, 1,2,3-trimethylbenzene, m-diethylbenzene |
| XYO | o-xylene, o-ethyltoluene |
| XYP | m-p-xylene, p-ethyltoluene, p-diethylbenzene |
| HCHO | formaldehyde |
| ACD | acetaldehyde |
| MVK/MACR | methyl vinyl ketone and methacrolein |

**Table 4.** Isoprene oxidation mechanism replacing the isoprene chemistry in RACM 2.

| reaction | reaction rate constant / $\mathrm{cm^3 s^{-1}}$ | reference |
|---|---|---|
| ISOP–>MACR+HCHO+OH | $0.31 \times 1.8 \times 10^{11} \times \exp(-9752/T)$ | a |
| ISOP–>MVK+HCHO+OH | $0.62 \times 1.04 \times 10^{11} \times \exp(-9746/T)$ | a |
| ISOP–> HPALD1+HO2+HPCARPO2 | $0.5 \times 0.62 \times (9.5 \times 10^{7} \exp(-7009/T) + 1.79 \times 10^{-7} \exp(3722.5/T) \times k_{tr}{}^{f})$ | a |
| ISOP–> HPALD2+HO2+HPCARPO2 | $0.5 \times 0.31 \times (3.8 \times 10^{13} \exp(-10745/T) + 5.82 \times 10^{-2} \exp(476.3/T) \times k_{tr}{}^{f})$ | a |
| HPALD1+HV–>OH+HO2+0.5×HKET +0.5×MGLY+0.5×ALD+HCHO | $100 \times j\mathrm{MACR}$ | b |
| HPALD2+HV–>OH+HO2+0.5×HKET +0.5×GLY+0.5×ALD+HCHO | $100 \times j\mathrm{MACR}$ | b |
| HPALD1+OH–>OH | $4.6 \times 10^{-11}$ | b |
| HPALD2+OH–>OH | $4.6 \times 10^{-11}$ | b |
| HPCARPO2–>CO+OH+OP2 | $0.1$ | a |
| HPCARPO2+NO–>NO2+MGLY+OH+OP2 | $2.9 \times 10^{-12} \exp(-300/T)$ | a |
| HPCARPO2+HO2–> OP2 | $7.5 \times 10^{-13} \exp(-700/T)$ | a |
| ISHP+OH–>IEPOX+OH | $1.9 \times 10^{-11} \exp(-390/T)$ | c |
| ISHP+OH–>0.7×ISOP+0.3×MACR+0.3×OH | $0.38 \times 10^{-11} \exp(-200/T)$ | c |
| IEPOX+OH–>IEPOXO2 | $5.78 \times 10^{-11} \exp(-400/T)$ | c |
| IEPOXO2+NO–>IEPOXO+NO2 | $2.54 \times 10^{-12} \exp(-360/T)$ | |
| IEPOXO2+HO2–>IEPOXO+OH+O2 | $0.074 \times 10^{-11} \exp(-700/T)$ | c |
| IEPOXO–>0.125×OH+0.825×HO2+0.251×CO +0.725×HKET+0.275×GLY+0.275×ALD +0.074×ORA1+0.275×MGLY+0.375×HCHO | $1 \times 10^{6}$ | c |
| MCP–>HKET+OH+CO | $2.9 \times 10^{7} \exp(-5297/T)$ | d |
| MACP+NO–>0.65×MO2+0.65×CO +0.35×ACO3+NO2+HCHO | $2.54 \times 10^{-12} \exp(-360/T)$ | d |
| MCP+NO–>NO2+HO2+HKET+CO | $2.54 \times 10^{-12} \exp(-360/T)$ | d |
| MVKP+HO2–>OP2 | $0.34 \times 2.91 \times 10^{-13} \exp(-1300/T)$ | e |
| MVKP+HO2–>ACO3+OH+ALD | $0.48 \times 2.91 \times 10^{-13} \exp(-1300/T)$ | e |
| MVKP+HO2–>HO2+OH+ORA2 | $0.18 \times 2.91 \times 10^{-13} \exp(-1300/T)$ | e |

[a] Peeters et al. (2014)

[b] Peeters and Müller (2010)

[c] Paulot et al. (2009)

[d] Crounse et al. (2012)

[e] Praske et al. (2015)

[f] $k_{tr}$ = NO$\times 2.43 \times 10^{-12} \exp(-360/T) +$ HO2$\times 2.05 \times 10^{-13} \exp(-1300./T) +$ ACO3$\times 8.4 \times 10^{-14} \exp(-221/T) +$ MO2$\times 3.4 \times 10^{-14} \exp(-221/T)$

**Table 5.** Median values of measured species for morning and afternoon hours.

| | 06:00–10:00 | 12:00–16:00 |
|---|---|---|
| $j(O^1D) \, / \, 10^{-5}\,s^{-1}$ | 0.63 | 1.3 |
| $j(NO_2) \, / \, 10^{-3}\,s^{-1}$ | 3.5 | 4.9 |
| $OH \, / \, 10^6\,cm^{-3}$ | 3.8 | 6.9 |
| $HO_2 \, / \, 10^8\,cm^{-3}$ | 1.9 | 7.4 |
| $RO_2 \, / \, 10^8\,cm^{-3}$ | 3.2 | 8.8 |
| $k_{OH} \, / \, s^{-1}$ | 20 | 11 |
| NO /ppbv | 2.5 | 0.25 |
| $NO_2$ /ppbv | 12 | 3.3 |
| HONO /ppbv | 0.78 | 0.51 |
| $O_3$ /ppbv | 39 | 93 |
| CO /ppmv | 0.70 | 0.54 |
| $CH_4$ /ppmv | 2.2 | 2.0 |
| ISO / ppbv | 0.59 | 0.84 |
| ETH / ppbv | 4.1 | 2.7 |
| HC3 / ppbv | 4.0 | 2.0 |
| HC5 / ppbv | 2.5 | 1.0 |
| HC8 / ppbv | 0.57 | 0.22 |
| ETE / ppbv | 3.3 | 0.93 |
| OLI / ppbv | 0.25 | 0.20 |
| OLT / ppbv | 0.83 | 0.21 |
| BEN / ppbv | 1.3 | 0.71 |
| TOL / ppbv | 1.6 | 0.69 |
| HCHO / ppbv | 8.4 | 7.5 |
| ACD / ppbv | 2.6 | 1.9 |
| MACR / pbv | 0.36 | 0.28 |
| MVK / ppbv | 0.54 | 0.43 |

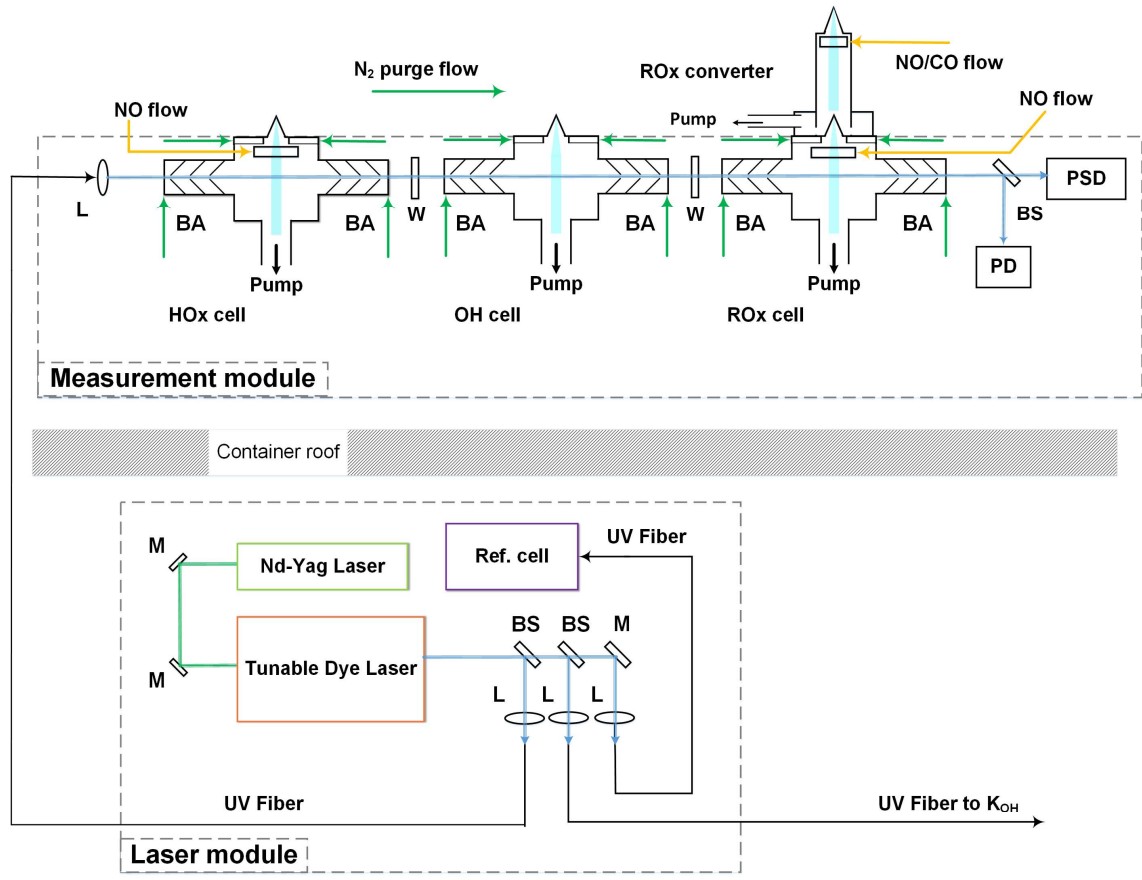

**Figure 1.** Schematic drawing of the LIF instrument for the detection of OH, $HO_2$ and $RO_2$. The laser module and the measurement module were installed inside and on top of a sea container, respectively. 308 nm laser-light is split into three parts (BS: beam splitter, L: lens) and guided by optical fibers to the measurement cells, the $k_{OH}$ instrument and the reference cell. Ambient air is sampled into low pressure fluorescence cells that are separated by windows (W). Reactive gases (NO, CO) are added into the $HO_x$- and $RO_x$- cells and the $RO_x$ converter. Baffle arms (BA) and fluorescence cells are continuously purged with $N_2$. The position and the power of the laser beam are monitored by a photodiode (PD) and a position-sensitive diode (PSD).

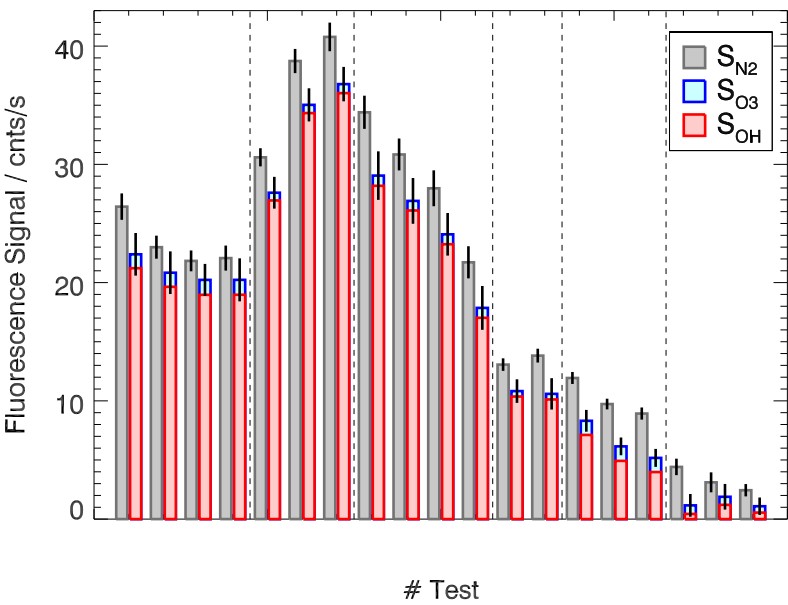

**Figure 2.** Results of chemical OH modulation tests performed during the campaign. In each test, the total measured OH signal without OH scavenger ($S_{N2}$) is compared to the sum of the known contributions from ambient OH ($S_{OH}$) and the interference from $O_3$ ($S_{O3}$). The error bars denote the $1\sigma$ statistical error. A fluorescence signal of 30 cnts/s (counts per s) corresponds to an OH concentration of $1.0\times10^7\,\mathrm{cm}^{-3}$.

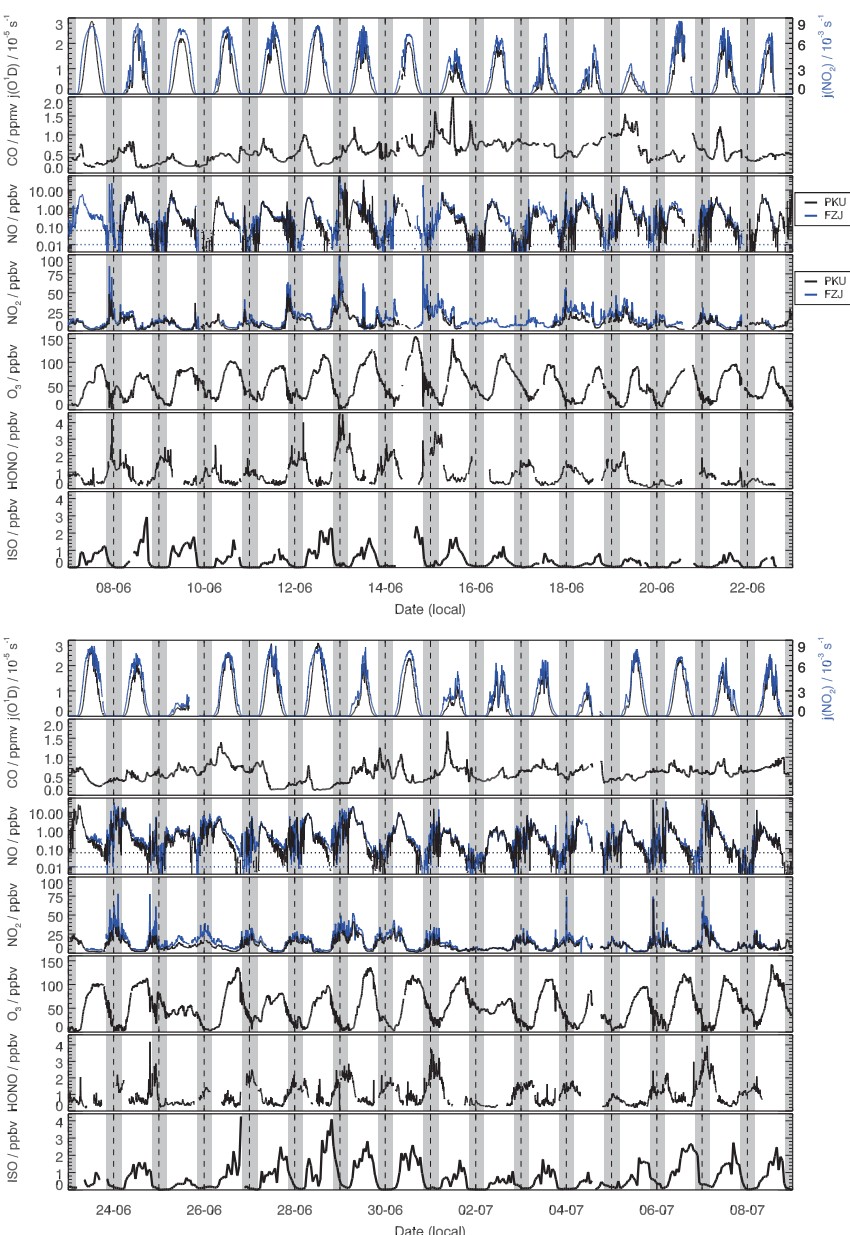

**Figure 3.** Time series (5-minute data) of measurements during this campaign for $j(O^1D)$, $j(NO_2)$, CO, NO, $NO_2$, HONO, $O_3$ and isoprene (ISO) used as constraints for model calculations. Vertical, dashed lines denote midnight. Grey areas indicate nighttime. Several species were measured by two instruments provided by PKU and FZJ. Measurements of both instruments for $O_3$ and CO well agreed, so that data sets were combined to close data gaps. Only the combined data set is shown here, but different colors indicate the origin of data. $NO_2$ and NO mixing ratios measured by the PKU instruments were generally 20 % smaller than those measured by the FZJ instrument. The horizontal lines denote the limit of detection for two NO instruments (10 pptv for FZJ; 60 pptv for PKU;). Both time series are shown, but measurements from the PKU instruments were used as model constraints.

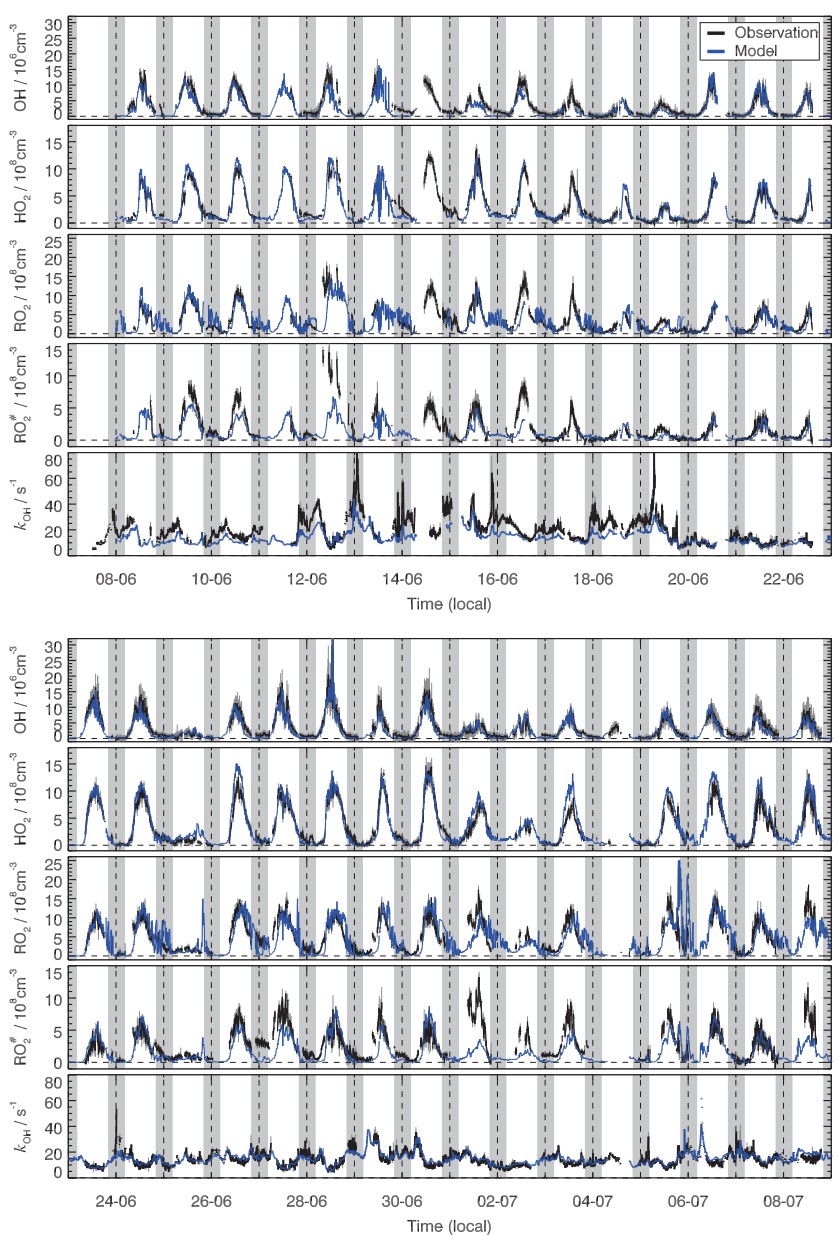

**Figure 4.** Time series of measured and modelled OH, HO$_2$, RO$_2^{\#}$, total RO$_2$ concentrations and $k_{\text{OH}}$. Vertical, dashed lines denote midnight. See text for details of the definition of RO$_2^{\#}$ and total RO$_2$. Grey vertical lines denote $1\,\sigma$ standard deviation for measured radicals concentration with respect to $5\,\text{min}$ variability. Grey areas indicate nighttime.

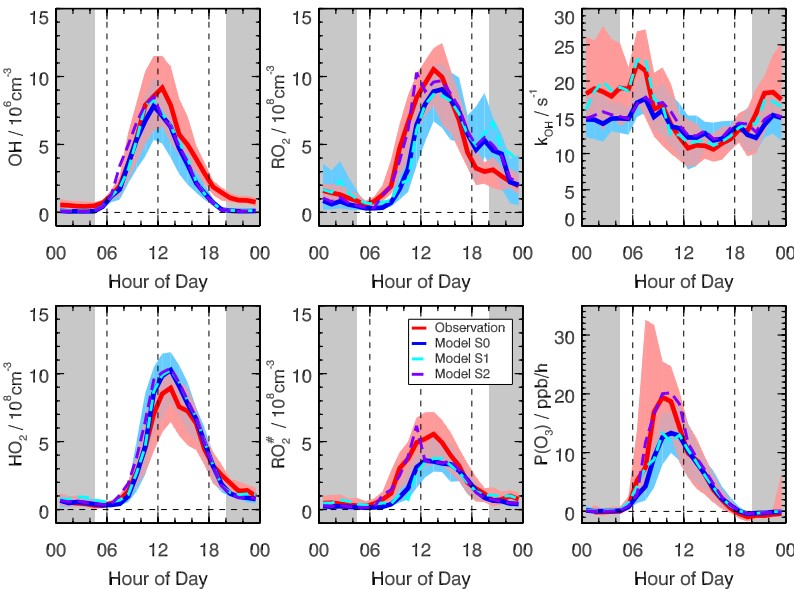

**Figure 5.** Comparison of hourly median diurnal profiles of OH, $HO_2$, $RO_2$, $RO_2^\#$ concentrations and $k_{OH}$ and the ozone production rate $P(O_3)$ (thick lines give median values, colored areas give 25 % and 75 % percentiles). S0 denotes results from the base model run. S1 shows results, when the VOC concentrations in the model are increased to match the observed OH reactivity. S2 shows results, when an additional primary $RO_2$ source ($2\,\mathrm{ppbv\,h^{-1}}$) is added in the model for the time between 6:00 and 12:00. Grey areas indicate nighttime.

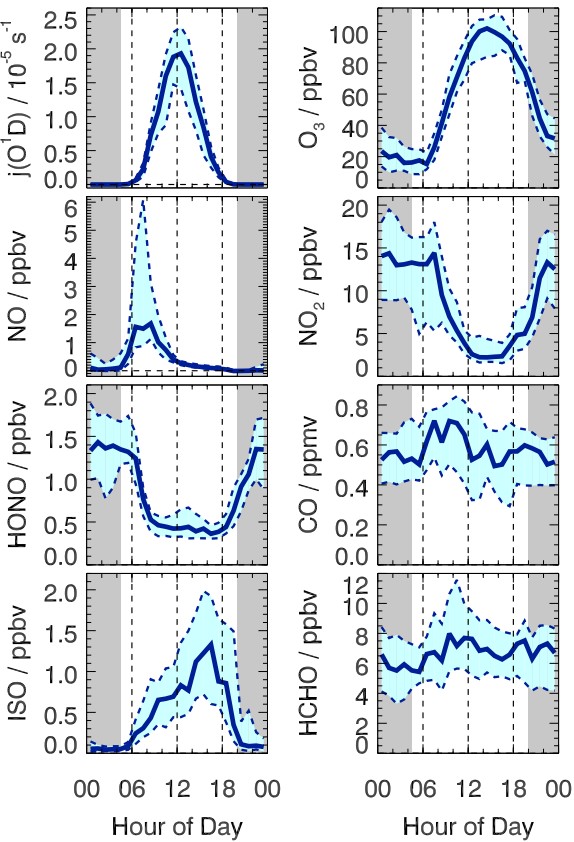

**Figure 6.** Hourly median diurnal profiles of measured j(O$^1$D), O$_3$, NO, NO$_2$, HONO, CO, isoprene (ISO) and HCHO (thick lines give median values, colored areas give 25 % and 75 % percentiles). Grey areas indicate nighttime.

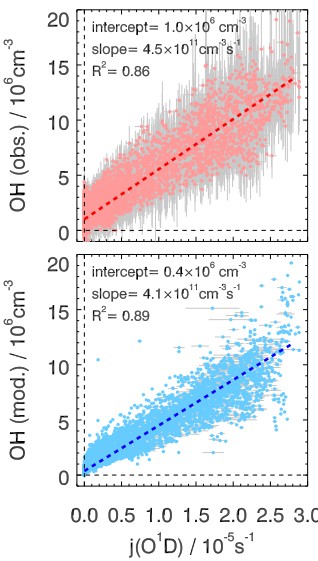

**Figure 7.** Correlation between $j(O^1D)$ and measured (upper panel) and modelled OH (lower panel). A linear fit is applied, which takes errors in both measurements into account.

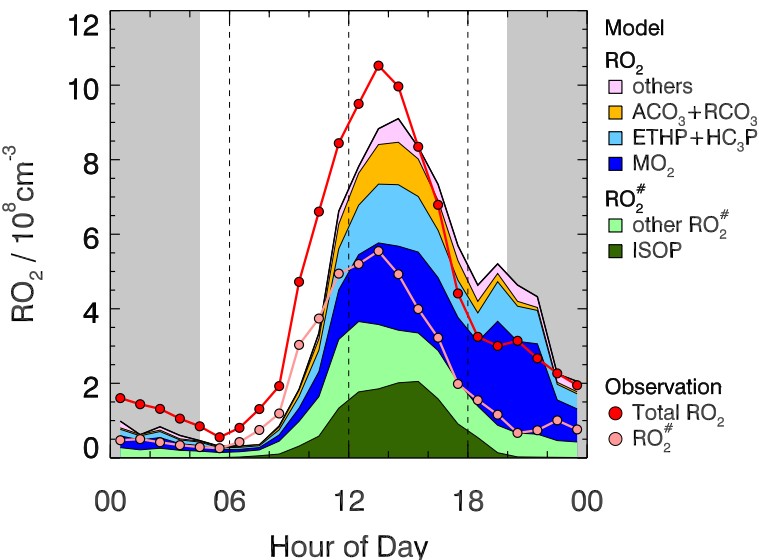

**Figure 8.** Hourly median diurnal profiles of measured and modelled $RO_2$ concentrations. Measurements can distinguish between total $RO_2$ concentrations and the subclass of $RO_2^{\#}$. Modelled $RO_2$ species are shown as colored areas. $MO_2$ are methyl peroxy radicals. ETHP are ethyl peroxy radicals. $HC_3P$ are alkyl peroxy radical (carbon number = 3 or 4). $ACO_3 + RCO_3$ are acetyl peroxy radicals. In the evening, "other" $RO_2$ radicals are mainly $RO_2$ species produced by the reaction of VOCs with $NO_3$. ISOP are isoprene peroxy radicals. "other" $RO_2^{\#}$ include peroxy radicals from long alkanes, alkenes, aromatics, and isoprene oxidation products (MVK and MACR). Grey areas indicate nighttime.

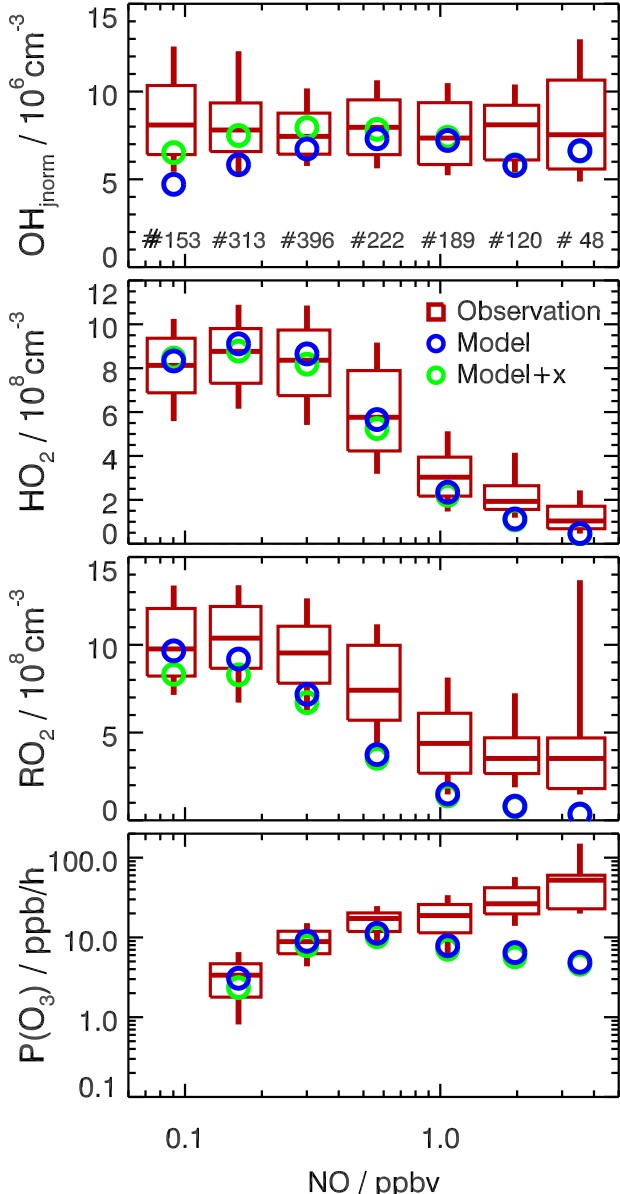

**Figure 9.** NO dependence of OH, $HO_2$ and $RO_2$ concentrations and instantaneous ozone production rate $(P(O_3)_{net})$ for daytime conditions $(j(O^1D) > 0.5 \times 10^{-5}\,s^{-1})$. OH concentrations are normalized to the average of $j(O^1D)$ $(1.5 \times 10^{-5}\,s^{-1})$. Boxes give 75 % and 25 % percentiles, the center lines the median, and vertical lines 90 % and 10 % percentiles for NO intervals of $\Delta\ln(NO)/ppbv=0.57$. Numbers in the upper panel give the number of data points included in the analysis of each NO interval. Only median values are shown for model results. Results from the base model and with additional radical recycling by a species X (equivalent to 100 pptv NO) are plotted.

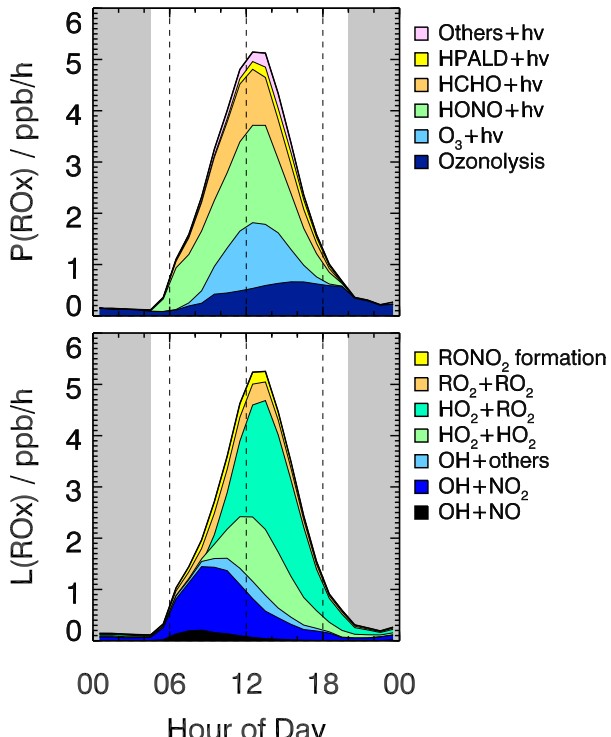

**Figure 10.** Hourly median diurnal profiles of modelled rates of primary $RO_x$ production and termination reactions. Grey areas indicate nighttime.

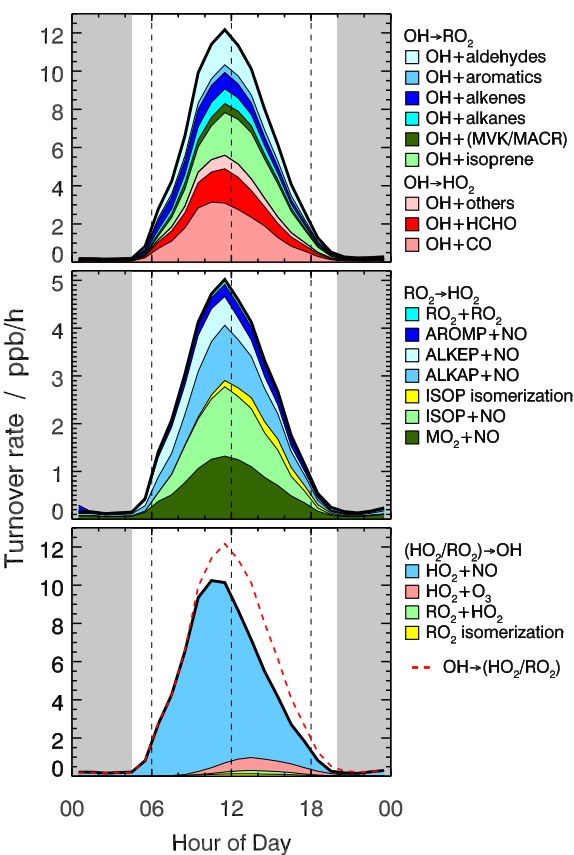

**Figure 11.** Hourly median diurnal profiles of turnover rates (model results) of radical propagation reactions between OH, $HO_2$ and $RO_2$ radicals. ALKAP: alkanes derived peroxi radicals; ALKEP: alkenes derived peroxi radicals; AROMP: aromatics derived peroxi radicals. Grey areas indicate nighttime.