# Peer review of "Radical chemistry at a rural site (Wangdu) in the North China Plain: Observation and model calculations of OH, HO2 and RO2 radicals"

_Atmospheric Chemistry and Physics, 2016_

## Referee Comment (RC1) · Anonymous Referee #1 · 22 Aug 2016

The manuscript reports HOx measurements and model comparison for the Wangdu region in China. The model is able to reproduce the daytime radical observations with two notable exceptions: OH concentrations are under-predicted at NO concentrations less than 300 pptv and RO2 radicals are under-predicted at high NOx concentrations which has implications for estimations of ozone production rates. The paper is well written and presents a balanced discussion of the findings in the context of previous results and is suitable for publication in Atmospheric Chemistry and Physics. I have a few comments below which should be addressed before final publications to strengthen the overall conclusions.

Interferences in the OH detection, OH chemical modulation tests: Was the propane

concentration varied in the field? From lines 265 – 267 it is unclear if the variable titration efficiency is a laboratory result or field observation. If it was the latter, how was titration efficiency determined in the field? Were laboratory tests conducted to ensure no internal removal of OH in the cell (line 273)? Figure 2: What do the dashed lines correspond to? It is difficult to assess from this figure if there is any diurnal variation in the magnitude of the interference signal? Could the authors comment on any variation observed, e.g. as a function of atmospheric composition? Is this possible unknown interference signal of sufficient magnitude to account for the modelled measured OH discrepancy at [NO] < 300pptv (line 479)?

Possible RO2 interference: Fuchs et al. (Review of Scientific Instruments, 2008) report a possible interference in the RO2 instrument from pernitric acid and methyl peroxy nitrate which have the potential to thermally decompose in ROx system and be detected as HO2 and CH3O2. Could the authors comment on the impact this interference may have for these field conditions, particularly under the high NOx conditions experienced in the morning? Could this interference explain the model measured discrepancy in RO2 at this time? What is the impact of this interference on the ozone production rate calculated from the measured HO2 and RO2 concentrations?

Model measurement comparison of RO2: The manuscript focusses on the differences observed between measured and modelled RO2 in the morning, but in figure 5 the model under-predicts RO2 and RO2# until ∼16:00. Some comments should be provided on this under-prediction; the under-prediction in OH reactivity cannot account for this under-prediction beyond 10am. Please extend this commentary to lines 849 in the Conclusion also. Owing to the strong coupling between RO2, HO2 and OH (highlighted in figure 11), how does the model under-prediction of total RO2 impact the model's ability to predict OH and HO2? Could the model be scaled to reproduce [RO2] and then the performance of the model to predict OH and HO2 re-assessed? There are inconsistencies in the modelled and measured radical ratios (and OH reactivity) that warrant further investigation. Section 3.6 would benefit from a more detailed discussion of the

modelled RO2 species – what are the other RO2# species in figure 8? Please define 'MO2' in this figure caption.

Minor comments:

Line 331: Do the RO2# concentrations determined in the HO2 and RO2 cell agree?

Line 358: Please give the typical solar background at noon. Was it necessary to shade the cells?

Line 527: 'HO2 concentrations are well reproduced by the model during the daytime'. From figure 5 it looks like the model has a tendency to over-predict [HO2] in the afternoon. Could this over-prediction be masking the full magnitude of the model under-prediction of OH at this time (as HO2 is a strong secondary source of OH)?

Line 593-594: 'scaling VOC concentrations to match measurements..' which VOC species were scaled? Does the VOC species chosen influence the modelled [RO2]?

Line 595 ' can be partly closed..' and also, line 701 'rate better agrees..' please provide the percentage change.

Line 720 – 723: This statement seems to be at odds with the model-measurement comparison presented in this manuscript which shows good agreement between modelled and measured HO2 in the morning but a modelled measurement discrepancy for RO2. Calculating ozone production from an RO2 concentration estimated from HO2 could mask a high morning ozone production rate.

Line 769: There is no experimental evidence that HONO formed from the reaction of HO2.H2O + NO2, as postulated by Li et al., (2014), occurs and so shouldn't be speculated on here.

Line 770: Could an example of OH+hydrocarbon which does not form HO2 or RO2 be provided here.

Line 848: Please provide the equivalent NO required in the previous campaigns for

comparison.

---

## Referee Comment (RC2) · Anonymous Referee #2 · 27 Aug 2016

This paper presents measurements of OH, HO$_2$ and RO$_2$ radicals using laser-induced fluorescence in a rural site in China together with box model calculations. The authors find that the model predicted radical concentrations are in reasonable agreement with the observations when mixing ratios of NO were greater than 1 ppbv, similar to previous measurements in urban environments. However they find that the model underestimates the observed OH concentrations when mixing ratios of NO were less than 300 pptv. The authors performed some tests to determine whether unknown interferences contributed to the measured OH concentrations, and find that for the limited number of tests performed the measured interference cannot explain the discrepancy between the model predictions and the measurements. Including an unknown species

that recycles OH equivalent to 100 pptv NO brings the model into better agreement with the measured OH concentrations. The model also underestimates the observed $RO_2$ concentrations during the morning when NO is high, resulting in the model underestimating the instantaneous rate of ozone production. Increasing the OH reactivity by VOCs to match the observed reactivity improves the agreement of the modeled $RO_2$ concentrations with the model.

The paper is reasonably written but could use proofreading to improve English grammar and punctuation. The paper would be acceptable for publication in ACP after the authors have addressed the following comments.

1) The authors performed several Interference measurements using an external chemical titration technique. Unfortunately it appears that these interference measurements were not done continuously but were done only on four specific days. However, it is not clear exactly when the tests were done and what the ambient conditions were during each test. Were any tests done when NO was less than 300 pptv, the conditions when the model-measurement discrepancies were the greatest, or was the measured interference similar for all ambient levels of NO? This should be clarified. Adding the times when these tests were done to Figure 3 would provide more information on whether these tests were done under typical ambient conditions for the campaign. Was this interference subtracted from all of the OH measurements?

2) On page 17 the authors state that the measured OH concentrations are approximately $1 \times 10^6$ $cm^{-3}$ greater than model predictions during the afternoon when the mixing ratios of NO decrease from 0.3 to 0.1 ppb. This discrepancy appears to be consistent with the average measured interference of $1 \times 10^6$ $cm^{-3}$ described on page 14, suggesting that the observed discrepancy with the model could be due to the interference. This possibility should be discussed in more detail.

3) In their measurements of $HO_2$, the authors varied the added NO to determine the interference from alkene and aromatic peroxy radicals. However, it is unclear to me how

the authors determined the $RO_2$ conversion efficiencies described on page 11 unless the absolute conversion efficiency for one of the NO flows was determined through calibrations with known concentrations of peroxy radicals. Did the authors perform $RO_2$ conversion efficiency calibrations similar to that described in Fuchs et al., 2011? This should be clarified.

4) It is not clear how the authors derive the $RO_2\#$ concentrations and compare it to the model. The measured $HO_2^*$ in the $RO_x$ channel reflects the conversion of alkene, aromatic, and other $RO_2$ radicals to $HO_2$ in the detection cell with a conversion efficiency dependent on the $RO_2$ radical as described in Fuchs et al. (2011). Subtracting the $HO_2$ measured in the $HO_2$ axis gives $\alpha_\# RO_2\#$. Ideally, the authors should compare this measured value which is the result of various conversion efficiencies to the modeled $RO_2\#$, where the individual modeled $RO_2$ concentrations are scaled by their expected conversion efficiencies, which are not necessarily all 0.8. However, it appears that the authors are scaling the measured $RO_2\#$ by an average conversion efficiency of 0.8 and comparing this value to the modeled concentration of the sum of the interfering $RO_2$ concentrations. This should be clarified. Have the authors measured the individual $RO_2$ conversion efficiencies for their instrument?

5) The authors state that the underestimation of the $RO_2$ concentrations by the model during the high NO conditions in the morning is improved when the OH reactivity of the model is increased, but few details are provided. Similar results were found during CalNex by Griffith et al. (JGR, 2016). How much did the modeled $RO_2$ increase in this scenario? Perhaps the results of this model run could be added to Figures 5 and 9.

6) Similarly, the authors find that the model underestimates the rate of ozone production under high NO conditions due to the underestimation of $RO_2$ radicals by the model. Similar results were found during CalNex (Brune et al., Faraday Discuss., 2016, 189, 169; Griffith et al., JGR, 2016). Does the underestimation of $RO_2$ (and therefore $PO_3$) depend on the measured OH reactivity? Griffith et al. (2016) found that the underestimation of $PO_3$ by the model was higher when the OH reactivity from VOCs was the

greatest.

---

## Referee Comment (RC3) · Anonymous Referee #3 · 12 Sep 2016

**Review of "Radical chemistry at a rural site (Wangdu) in the North China Plain: Observations and model calculations of OH, HO$_2$ and RO$_2$ radicals" by Tan et al.**

This paper summarizes measurements made in China in 2014 as part of a comprehensive field campaign that included measurements of HOx radicals and many other controlling species and parameters. The observations are analyzed using a 0-dimensional box model with a modified RACM 2 chemical mechanism. Comparison of observations and model results yields specific instances of agreement and some situations of significant disagreement. The authors hypothesize reasons for those differences that could lead to further research on the abundance of HOx radicals and their chemistry in the lower troposphere.

This reviewer believes that this is a fairly well-written and well-organized paper. The logic and presentation were mostly easy to follow, with a couple of minor exceptions. The paper places the current state of the art in HOx measurements in the context of our understanding of tropospheric chemistry for the environmental conditions encountered. There are no suggestions for major changes to the paper by this reviewer, but there are suggestions for changes to grammar, and some questions on the approach and intent of the discussion.

Line 3. Suggest "Observation of radicals by the laser induced fluorescence (LIF technique revealed…"

Lines 7-8. Suggest "…the model can reproduce measured radical concentrations reasonably well during daytime."

Line 8. Suggest "As in previous…"

Line 14. It is a bit confusing to say the RO$_2$ is in good agreement, but then to say that RO$_2$ is underestimated by a large amount in the morning. Suggest changing this text to make the points clearer.

Line 23. Suggest "…has become an issue of great concern for citizens and…"

Line 24. Suggest "…strategies have already been implemented…"

Line 28. Suggest "…air quality has been steadily deteriorating in some locations…"

Line 45. Do the values of 0.8 and 0.4 ppbv refer to average or median values required? If so, state this.

Line 70. Misspelling of "formaldehyde".

Line 75. Suggest "…calculations compared to results from previous campaigns…"

Line 84. Suggest "…botanical garden…"

Line 85. Suggest "…were growing to within 10 m of the instruments."

Line 86. Suggest "There was no car or truck traffic in the botanical garden; The closest road was 2 km away."

Line 88. Suggest removing "from the other containers"

Line 97. Suggest "…monitored by multiple instruments."

Line 109. Note that peroxy radicals will also shift the NO to NO$_2$ ratio as the air travels down the sample line.

Line 113. Suggest "…reproducible (10 %) than in previous deployments…"

Table 1.  All of the various techniques when multiple instruments were measuring are not given. Suggest making the list complete.

Line 120.  This discussion of HONO measurements is good, but given the potential uncertainties in such observations, it might be good to do a more detailed comparison, perhaps including a figure comparing all six measurements.

Line 127.  Suggest another phrase rather than "generally reasonable" such as "within quoted errors".

Line 130.  An instrument with higher sensitivity is more sensitive.  This is a common confusion.  Suggest changing to "are generally better,"

Line 152.  Suggest "A 20 m high tower with meteorological instrumentation was set up 15 m south of the containers, where…"

Line 176.  Suggest "…similar to instruments from this organization that have been described earlier…"

Line 201.  Suggest "…used to detect fluorescence photons…"

Line 214.  Suggest "…laser wavelength is tuned off the absorption line."

Line 219.  Suggest "…parameterized using the laser power, and the $O_3$ and water vapour concentrations."

Line 220.  It says that the correction is small compared to ambient OH, but this depends on the conditions.  Near sunrise and sunset (or at night), this could be a large correction.

Line 225.  It says that there was no interference from ozonolysis of simple alkenes, but what about larger, more complex alkenes (non-biogenic) that could be present?

Lines 245-246.  Suggest "…chemical modulation that was temporarily attached to the OH detection cell during selected periods."

Line 252.  Suggest "…it showed instabilities causing…"

Line 253.  Suggest starting a new paragraph on "The two signals…"

Line 257.  This reviewer does not like the use of "titration" in this context.  The authors can do as they choose, but suggest using "removal" or "conversion" efficiency rather than titration.   Also suggest removing both commas on this line.

Line 262.  Suggest "…that is measured with no OH scavenger added ($S_{N2}$)."

Line 267.  Suggest removing comma.

Line 268.  Suggest removing "flow".  Suggest "A flow of 0.02 to 0.2 lpm of a 5 % propane…"  Suggest using "lpm" rather than "Liter/min" throughout.

Line 271-272.  The issue of mixing reagents into a flow containing $HO_x$ radicals at ambient pressure has been solved by others, particularly those make CIMS-based $HO_x$ measurements (e.g. Mauldin et al.).

Line 277.  It states that the titration unit caused a 5% difference in OH sensitivity.  Was this applied to the data collected while it was present?

Line 282.  Suggest removing comma.

Line 286.  Suggest removing "cell".

Line 287.  Suggest "…which is comparable to the detection limit."

Line 294.  Suggest "…with an average conversion…"

Line 300.  Suggest removing second comma

Line 301.  Suggest "At the beginning…"

Line 307.  It states that it is assumed that the contribution of $RO_2^{\#}$ scales with added NO.  Is this justified by lab studies?  How are the correction factors given in lines 309-310 applied? ($HO_2 = HO_2^*/CF$ ?)

Line 319.  Suggest "… these are the dominate $RO_2$…"

Line 325.  Suggest removing comma.

Line 331.  Suggest "An average conversion efficiency…"

Line 333.  Suggest "Any error in this value adds…"

Lines 334-337.  Are the ambient data corrected for the artifacts as described?

Line 344.  Suggest stating the reagents used to convert OH to $HO_2$ and $CH_3O_2$ ($CO$ and $CH_4$?).

Lines 345-346.  Suggest "No trends with time for any of the sensitivities were observed."

Lines 349-350.  Suggest "…for the $HO_2$ cell, 5% and 10% for the high and lower $HO_2$ conversion efficiencies, respectively."

Line 358.  Based on the discussion, the detection limit at noon is about $1.5 \times 10^6$.  Is this a systematic or random effect of the solar light leakage?  Can the data be corrected for this?  Is this included in the overall measurement uncertainties?

Line 363.  Do the authors believe that RACM 2 is a suitable mechanism to study detailed $HO_x$ radical chemistry?  Why not use the explicit MCM mechanism, modified as you did to included updated isoprene chemistry?  Perhaps add some discussion as to why the RACM 2 mechanism was selected.

Line 385.  Suggest removing "equidistant"

Line 389.  When saying that the OH reactivity can "be well explained", suggest adding a quantitative value to the degree of agreement (within 22% or whatever).

Line 394.  Suggest replacing "much" with "very".

Line 398.  What species are being referred to as "these species"?  Suggest a bit more text to make it clear.

Lines 401-402.  Suggest "The largest class of $RO_2$ radicals that is not included in the …"

Line 425.  Suggest removing comma after "daytime".

Line 447.  Suggest "…after harvesting in surrounding…"

Line 452.  Suggest "…$NO_2$ often showed trends similar to CO…"

Line 455.  Suggest changing "often" to "sometimes".

Line 456.  Suggest changing "nitrogen monoxide" to "nitric oxide".

Line 464.  Suggest adding ", respectively" to the end of the sentence.  Suggest changing "small" to "low".

Line 465.  Suggest changing "raised" to "increased".

Line 467.  Suggest "…2 ppbv, leading to enhanced OH production from HONO photolysis."  One question: why didn't the peroxy radical concentrations also increase during this time period?

Lines 485-494. It appears to me that the $NO_3$ interference is sufficient to explain some or all of the nighttime signal observed.

Lines 495-503. Suggest a discussion and perhaps a figure showing the major contributors to the OH reactivity.

Line 509. Suggest chaining "as" to "using".

Lines 515-516. Suggest rewording this sentence. One suggestion would be to separate the data into two equal groups rather than have this long discussion about why the two groups are not equal in size.

Line 522. The median measure-model difference discussed is of the order of the various artifacts and interferences. Have the data been corrected for all of them before doing this comparison? If so, suggest stating this somewhere.

Line 523. It is not clear what is meant by "At the same time". Suggest rewording to make this clearer.

Line 551. Suggest changing "taking" to "using".

Line 563. Is there any evidence of organic nitrites contributing to the enhancement of peroxy radicals in the morning? Photolysis of such species, if they exist, could contribute to the difference seen.

Line 592. It states that the production rate of $RO_2$ could be underestimated, but one should also consider that the loss rate of $RO_2$ could be overestimated somehow.

Line 594. It states that VOC concentrations are scaled to match measurements. Which measurements? Are they the VOCs or $k_{OH}$?

Line 598. Suggest "…predict $RO_2$ at night…"

Line 599. Suggest "…caused by values near the limit of detection…"

Line 600. Suggest "…concentrations that are used as constraints." A thought on nighttime chemistry: if there is $NO_3$ present, then the NO concentration should be very small unless the $NO_3$ production rate is very large. This is because of the rapid reaction between NO and $NO_3$. This could help with the modeling of nighttime chemistry.

Line 606. Suggest "…ppbv when other conditions controlling OH are constant."

Line 607. Suggest "Figure 9 shows the dependence of the measured and modelled radicals concentrations on the NO mixing ratio.

Line 612. Suggest replacing "with" with "at"

Line 619. Suggest "OH behavior similar to that shown in…"

Line 628. Suggest replacing "when" with "as".

Line 629. Suggest "At 3 ppbv NO, the modelled $RO_2$ concentration is less than $1 \times 10^8$ $cm^{-3}$, whereas the median measured $RO_2$ is $3.5 \times 10^8$ $cm^{-3}$."

Line 631. Suggest "…than the model values…"

Line 644. Suggest removing comma.

Line 660. Suggest "….allow the calculation of net ozone…"

Line 665 and 668. Often $P(O_3)$ refers to gross ozone production and $P(O_3)-L(O_3)$ is the net. Suggest perhaps indicating by using $P(O_3)_{net}$.

Line 692.  Suggest "Cl radicals may…"

Line 693.  Suggest "…with a maximum…"

Line 710.  It is good to see meteorology mentioned as important to observed $O_3$ concentrations.  This is likely very important in determining the diurnal cycle of ozone.  Suggest changing text to "…by the increase of the boundary layer depth."

Line 730.  Suggest removing first comma.

Line 738.  Suggest "…and HCHO concentrations as well as…"

Lines 758-769.  Suggest including Ye et al in the discussion of HONO budgets.

Line 770.  Suggest giving an example of a reaction of OH with VOCs that do not lead to peroxy radicals.

Lines 787-788.  Suggest adding the word "each" before "account" since it appears that NO reactions with each of the species accounts for 26% of the conversion (total of about 50%).

Line 824.  Suggest "…applying a laser induced fluorescence…"

Line 832.  Yes, the interference would be minor compared to the daytime maximum, but it could be very important at sunrise and sunset.

Line 834.  Suggest "setup" rather than "set-up".

Line 842 and 846.  Suggest "As" in place of "Like".

Line 860.  Suggest "…but caused by values of the modelled $HO_2$ that are too low…"

References.  The papers on $HO_x$ measurements are very Euro-centric.  Suggest adding some papers from US $HO_x$ measurement groups.

Figure 1.  The colors for $S_{N2}$ and $S_{OH}$ are very similar.  Suggest changing one of them to a very different color.

Figures 3 and 4.  Suggest changing x-axis title to "Date (local)".

Figures 3, 4, 5, 6, 8, and 10.  Is the gray period meant to signify nighttime?  If so, the authors should check this carefully.  It appears that there are photolysis processes (such as $O_3$) that occur after sunset (see Figure 10).

---

## Author Comment (AC1) · 25 Nov 2016

We would like to thank the reviewer for comments and questions which helped us to improve the manuscript. The reviewer comments are given below together with our responses and changes made to the manuscript.

**1.1 Comments:** *Interferences in the OH detection, OH chemical modulation tests: Was the propane concentration varied in the field? From lines 265 – 267 it is unclear if the variable titration efficiency is a laboratory result or field observation. If it was the latter, how was titration efficiency determined in the field?*

**Response:** The amount of propane and corresponding titration efficiency (changed to removal efficiency according to Referee #3) was optimized and determined in the field before each titration test. More detail information is provided in the revised manuscript.

**Change:** We added on Page9 L269: "The knowledge of ε is essential for an accurate quantification of potential interferences. The removal efficiency was tested and optimized in the field using the OH calibration device as a radical source. The value of ε was found to depend on the flow rates of the added gases (propane and nitrogen). Propane was added as a 5% mixture in nitrogen with a flow rate between 0.02 and 0.2lpm (Liter per minute) which was further diluted in a carrier flow of pure nitrogen (0.04 to 0.5lpm). The dependence of ε on the flow rates showed that mixing of the injected propane into the high flow of ambient air was inhomogeneous similar to results reported in Novelli et al. (2014). Because of technical difficulties with the flow regulation, the removal efficiency was re-determined before each ambient titration test. The values obtained for ε ranged between 80% and 97% with an accuracy of 10% (1 σ) at fixed nominal propane and nitrogen flows."

**1.2 Comments:** *Were laboratory tests conducted to ensure no internal removal of OH in the cell (line 273)?*

**Answer:** We did kinetic calculation to estimate the impact of propane addition on internal removal of OH in the cell, which show only minor impact as long as the removal efficiency is less than 100%.

**Change:** We revised the text from Page 10 Line 273 to Line 275: "Kinetic

calculations show that the added propane removes less than 0.3% of internally produced OH. The calculation assumes that the added propane is homogeneously mixed in the sampled air, yielding an expected OH lifetime which is larger than 0.1s and therefore much longer than the residence time (3ms) in the low-pressure detection cell. Therefore, the propane concentrations used in the chemical-modulation tests are not expected to influence possible OH interference signals."

**1.3 Comments:** Figure 2: What do the dashed lines correspond to? It is difficult to assess from this figure if there is any diurnal variation in the magnitude of the interference signal? Could the authors comment on any variation observed, e.g. as a function of atmospheric composition? Is this possible unknown interference signal of sufficient magnitude to account for the modelled measured OH discrepancy at [NO] < 300pptv (line 479)?

**Answer:** In Figure 2, the dashed lines separate one set of test from the other. To assess the possible variation of the interference signals, we summarized the conditions during the 6 titration experiments in a new table. We found no evidence that the unaccounted interference correlates with other measured species. The signal appears to be similar with values within the range of $1 \times 10^6$ cm$^{-3}$ in quite different chemical conditions during the campaign. For a large range of NO concentrations, the residual OH determined in the titration tests was similar. If an accounted signal of $1 \times 10^6$ cm$^{-3}$ was subtracted from the measured OH concentration, the observed-to-modelled ratio of OH would be reduced from 1.4 to 1.2 for NO<300ppt and from 1.9 to 1.5 for NO<100ppt.

**Change:** We added a discussion about the averaged unaccounted signal and model-measurement discrepancy comparison and revised the text from Line 522 to Line 524: "The median diurnal profiles of the measured and modelled OH concentrations agree within their errors of 10% (1σ) and 40%, respectively, from sunrise to midafternoon. When the median NO mixing ratio (cf. Fig. 6) drops gradually from 0.3ppbv to 0.1ppbv in the afternoon, a systematic difference evolves, with measured OH concentrations being approximately $1 \times 10^6$cm$^{-3}$ higher than the model calculations. The discrepancy is of similar magnitude as the averaged unexplained OH determined in the chemical modulation experiments (Table 2). Thus, the overall agreement for OH would improve, if the unaccounted signal was fully considered as an OH

measurement interference. However, the underestimation of OH would persist for low NO conditions if a potential unaccounted signal was subtracted. When NO concentrations are less than 100pptv, the observed-to-modelled OH ratio would be reduced from 1.9 to 1.5, indicating that an OH source would still be missing for low NO conditions."

2. ***Comments:*** *Possible $RO_2$ interference: Fuchs et al. (Review of Scientific Instruments, 2008) report a possible interference in the $RO_2$ instrument from pernitric acid and methyl peroxy nitrate which have the potential to thermally decompose in ROx system and be detected as $HO_2$ and $CH_3O_2$. Could the authors comment on the impact this interference may have for these field conditions, particularly under the high NOx conditions experienced in the morning? Could this interference explain the model measured discrepancy in $RO_2$ at this time? What is the impact of this interference on the ozone production rate calculated from the measured $HO_2$ and $RO_2$ concentrations?*

**Answer:** We calculated the thermal decomposition of the peroxy nitric acid ($HO_2NO_2$), methyl peroxy nitrate ($CH_3O_2NO_2$) and PAN as Fuchs et al. (2008) did. Although these species could cause interference and help to explain the underestimation of $HO_2$ and $RO_2$, the impact is so small that have only minor impact on our measurements for high NO condition.

**Change:** We added on Page 11 L338: 'A bias in the measurement of $RO_2$ may be caused in polluted air by peroxy radicals, which are produced in the low-pressure converter of the $RO_2$ instrument by thermal decomposition of peroxy nitric acid ($HO_2NO_2$), methyl peroxy nitrate ($CH_3O_2NO_2$) and PAN (Fuchs et al., 2008). In the atmosphere, $HO_2NO_2$ and $CH_3O_2NO_2$ are in a fast thermal equilibrium with $HO_2$ and $CH_3O_2$, respectively, together with $NO_2$. The possible interference scales with $NO_2$, which was highest during the Wangdu campaign in the morning (median value of 15 ppbv; cf. Fig. 6). For this condition, according to model calculations by Fuchs et al. (2008), $HO_2NO_2$ and $CH_3O_2NO_2$ are expected to produce interferences of +2.6 % and +9 % for the detected $HO_2$ and $CH_3O_2$ radicals, respectively. Since $HO_2$ and $CH_3O_2$ contributed about 50 % (measured) and 10 % (modelled) to the total ROx in the morning, the estimated

interference for measured $RO_2$ is only +2 %. The interference from PAN decomposition in the instrument was calculated by Fuchs et al. (2008) to be 0.1 pptv per ppbv of PAN. Since PAN concentrations modelled for the Wangdu campaign are less than 1 ppbv, also from this compound no significant interference is expected. Another bias may be due to the perturbation of the reactor chemistry from high ambient NO concentrations (Fuchs et al., 2008). For the measurements in the ROx and $HO_2^*$ mode, the corresponding interferences are estimated to be less than +1 % and +3 %, respectively, at 15 ppbv NO.'

3. **Comments:** *Model measurement comparison of $RO_2$: The manuscript focusses on the differences observed between measured and modelled $RO_2$ in the morning, but in figure 5 the model under-predicts $RO_2$ and $RO_2^\#$ until 16:00. Some comments should be provided on this under-prediction; the under-prediction in OH reactivity cannot account for this under-prediction beyond 10am. Please extend this commentary to lines 849 in the Conclusion also. Owing to the strong coupling between $RO_2$, $HO_2$ and OH (highlighted in figure 11), how does the model under-prediction of total $RO_2$ impact the model's ability to predict OH and $HO_2$? Could the model be scaled to reproduce $[RO_2]$ and then the performance of the model to predict OH and $HO_2$ re-assessed? There are inconsistencies in the modelled and measured radical ratios (and OH reactivity) that warrant further investigation. Section 3.6 would benefit from a more detailed discussion of the modelled $RO_2$ species – what are the other $RO_2^\#$ species in figure 8? Please define 'MO_2' in this figure caption.*

**Answer:** Considering the uncertainties, $RO_2$ and $RO_2^\#$ were significantly underestimated only during the morning (06:00—10:00) hours. To resolve the issue of the model under-estimation of the $RO_2$ and $RO_2^\#$, three sensitivity studies were done to investigate the possible impact of a faster recycling (S1), a primary source (S2) and a slower removal rate (S3) of $RO_2$. In the revised manuscript, we added the results from two sensitivity tests to the Figure 5 (see following).

**Change:** In the revised manuscript, we added the sensitivity run with scaled VOCs

(S1) and additional primary source (S2) in Figure 5.

[Figure]

Figure 5 Comparison of hourly median diurnal profiles of OH, HO$_2$, RO$_2$, RO$_2^\#$ concentrations and k$_{OH}$ and the ozone production rate P(O$_3$) (thick lines give median values, colored areas give 25% and 75% percentiles). S0 (blue line) denotes results from the base model run. S1 (cyan, dashed line) shows results, when the VOC concentrations in the model are increased to match the observed OH reactivity. S2 (violet, dashed line) shows results, when an additional primary RO$_2$ source (2ppbvh$^{-1}$) is added in the model for the time between 6:00 and 12:00. Grey areas indicate nighttime.

We added a sentence at the end of Section 2.4 Page 13 Line 408: "The uncertainty of measurements and modelling needs to be taken into account in the comparison. The uncertainty of radical measurements is mainly determined by the measurement accuracies (OH: ±11%, HO$_2$: ±16%, RO$_2$: ±18%). A series of tests based on Monte Carlo simulations show that the uncertainty of the model calculations is approximately 40%."

We revised the text in Page 17 Line 528 "RO$_2$ and RO$_2^\#$ were significantly underestimated during the morning (06:00—10:00) hours with an observed-to-modelled ratio of 3 to 5, which is larger than the combined uncertainty (a factor of 2).

Reasons for discrepancies between measured and modelled $RO_2$ are further analyzed in Section 3.6."

Page 18 from Line 569 to Line 596, we revised the text "The strong underprediction of the observed $RO_2$ by more than a factor of 4 in the morning cannot be explained by the measurement errors and interferences discussed in Sections 2.3.4 and 2.3.5. In order to explore potential reasons for this underprediction, several sensitivity tests were performed. First, the impact of a faster OH to $RO_2$ conversion by an increased amount of VOC was tested (model sensitivity run S1). Second, an additional primary source of $RO_2$ was introduced into the chemical mechanism (S2). Third, the possibility of a slower removal rate of $RO_2$ was tested (S3).

[revised manuscript text omitted]

A more detailed discussion of the modelled $RO_2$ species is added in Sect. 3.6 as suggested. In Line 563 Page 18, we added: "In the group of modelled $RO_2^{\#}$ species, isoprene peroxy radicals (ISOP) make the largest contribution during daytime. Other modelled $RO_2^{\#}$ include peroxy radicals from alkenes, aromatics, long-chain ($> C_4$) hydrocarbons, and MVK and MACR. Among the $RO_2$ radicals which do not belong into the $RO_2\#$ group, peroxy radicals of short-chain ($<C_5$) alkanes are dominating: methyl peroxy radicals (MO2), ethyl peroxy radicals (ETHP), and peroxy radicals $HC_3P$ from $HC_3$ (e.g., propane). Acetyl peroxy radicals ($ACO_3+RCO_3$) are also a substantial fraction of $RO_2$."

**Minor comments:**

**Comments:** *Line 331: Do the $RO_2^{\#}$ concentrations determined in the $HO_2$ and $RO_2$ cell agree?*

**Answer:** $RO_2^{\#}$ is only measured in the $RO_2$ cell. The $HO_2$ cell measured $HO_2$ with a

minimum contribution from $RO_2^{\#}$ (see section 2.2.3).

**Comments:** *Line 358: Please give the typical solar background at noon. Was it necessary to shade the cells?*

**Answer:** We revised the text in 358: 'The typical solar background was about 40 cts/s which is a factor of 20 higher than the typical background signals obtained at night. Therefore, the detection limit was reduced by a factor of 5. A shade-ring was installed during the campaigns to shield the cell from direct solar radiation.'

**Comments:** *Line 527: 'HO$_2$ concentrations are well reproduced by the model during the daytime'. From figure 5 it looks like the model has a tendency to over-predict [HO$_2$] in the afternoon. Could this over-prediction be masking the full magnitude of the model underprediction of OH at this time (as HO$_2$ is a strong recycling of OH)?*

**Answer:** In a sensitivity model run, we constrained the $HO_2$ concentration to the observed values and found the underprediction of OH would be more significant, because the original model predicted slightly higher concentrations of $HO_2$. The observed-to-modelled ratio of OH would increase to 1.8 in this case. An analysis of the OH budget using only measurements is done in our accompanying paper by Fuchs et al. 2016. This study shows that the OH destruction cannot fully be explained by known production rates in the late afternoon, when NO concentrations are lowest. We revised the text in line 527:"In general, $HO_2$ concentrations are reproduced by the model during daytime within the combined uncertainties of measurements and model calculations. Nevertheless, the model has a tendency to over-predict $HO_2$ in the afternoon. If we constrain the model to the observed $HO_2$ concentrations, the observed-to-modelled OH ratio increases from 1.6 to 1.8 for daytime averaged conditions (04:30–20:00)."

**Comments:** *Line 593-594: 'scaling VOC concentrations to match measurements.' which VOC species were scaled? Does the VOC species chosen influence the modelled [RO$_2$]?*

**Answer:** All input VOCs are scaled (see in table 2) to remain the same ratio between

different VOC species. Sensitivity tests showed that the modelled $RO_2$ is not very sensitive to different specific VOC species since the required change of modeled $k_{OH}$ is relatively small (< 20%). The detailed explanation described above was added in the discussion of the $RO_2$ underprediction (refer to question 3).

**Comments:** *Line 595 'can be partly closed.' and also, line 701 'rate better agrees..' please provide the percentage change.*

**Answer:** More detail information is provided in the text as: "The observed-to-modelled $RO_2$ ratio is improved from 2.8 to 1.7"

**Comments:** *Line 720 – 723: This statement seems to be at odds with the model-measurement comparison presented in this manuscript which shows good agreement between modelled and measured $HO_2$ in the morning but a modelled measurement discrepancy for $RO_2$. Calculating ozone production from an $RO_2$ concentration estimated from $HO_2$ could mask a high morning ozone production rate.*

**Answer:** The statement is changed to: 'Total photochemical ozone production rates were directly measured in a sunlit environmental chamber during the SHARP campaign in Houston (Texas) 2009 (Cazorla et al., 2012; Ren et al., 2013). The comparison with ozone production rates determined from measured $HO_2$ and from modelled $HO_2$ and $RO_2$ suggests that the model underestimated both $HO_2$ and $RO_2$ at high NOx in the morning.'

**Comments:** *Line 769: There is no experimental evidence that HONO formed from the reaction of $HO_2.H2O + NO_2$, as postulated by Li et al., (2014), occurs and so shouldn't be speculated on here.*

**Answer:** We canceled the statement as suggested.

**Comments:** *Line 770: Could an example of OH+hydrocarbon which does not form $HO_2$ or $RO_2$ be provided here.*

**Answer:** We modified the text in line 770 as "Further radical terminating OH losses include reactions with unsaturated dicarbonyls (DCB1, DCB 2, DCB 3) and acetyl nitrate species (PAN, MPAN, etc) in RACM2."

**Comments:** *Line 848: Please provide the equivalent NO required in the previous campaigns for comparison.*

**Answer:** The equivalent NO required in the previous campaign is 0.8ppbv for PRIDE-PRD2006 and 0.4ppbv for CAREBEIJING2006. We rephrased the text: "…This behaviour is qualitatively in agreement with previous results from two field campaigns in China, in the Pearl River Delta and in the North China Plain, where the required equivalent NO is 800pptv and 400pptv (Lu et al., 2012, 2013)."

---

## Author Comment (AC2) · 25 Nov 2016

We would like to thank the reviewer for comments and questions which helped us to improve the manuscript. The reviewer comments are given below together with our responses and changes made to the manuscript.

1) **Comments:** *The authors performed several Interference measurements using an external chemical titration technique. Unfortunately it appears that these interference measurements were not done continuously but were done only on four specific days. However, it is not clear exactly when the tests were done and what the ambient conditions were during each test. Were any tests done when NO was less than 300 pptv, the conditions when the model-measurement discrepancies were the greatest, or was the measured interference similar for all ambient levels of NO? This should be clarified. Adding the times when these tests were done to Figure 3 would provide more information on whether these tests were done under typical ambient conditions for the campaign. Was this interference subtracted from all of the OH measurements?*

**Answers:** Because the titration device was a prototype, the chemical modulation tests were only performed on four days. The time period and chemical conditions are summarized and showed in a new figure in the revised manuscript. The unexplained residual signal are constant in the range of $1 \times 10^6 \text{cm}^{-3}$, independent on the ambient NO concentrations.

Referee #1 also had similar questions about the titration experiments. Therefore, we would like to refer to our answer to question 1 of Referee #1 for more detail information.

**Change:** We added a sentence in Line 431: "Because the test results are not sufficiently accurate to draw firm conclusions about an unknown interference, the OH data in this work was not corrected for a potential interference. Instead, the differences found in Fig. 2 are treated as an additional uncertainty of the OH measurements presented in this paper."

2) **Comments:** *On page 17 the authors state that the measured OH concentrations are approximately $1 \times 10^6 \text{ cm}^{-3}$ greater than model predictions during the afternoon when the mixing ratios of NO decrease from 0.3 to 0.1 ppb. This discrepancy appears*

*to be consistent with the average measured interference of $1 \times 10^6 cm^{-3}$ described on page 14, suggesting that the observed discrepancy with the model could be due to the interference. This possibility should be discussed in more detail.*

**Answers:** If an accounted signal of $1 \times 10^6$ cm$^{-3}$ was subtracted from the measured OH concentration, the observed-to-modelled ratio of OH would be reduced from 1.4 to 1.2 for NO < 300ppt and from 1.9 to 1.5 for NO < 100ppt.

**Change:** We added a discussion about the averaged unaccounted signal and model-measurement discrepancy comparison and revised the text from Line 522 to Line 524: "The median diurnal profiles of the measured and modelled OH concentrations agree within their errors of 10% (1σ) and 40%, respectively, from sunrise to midafternoon. When the median NO mixing ratio (cf. Fig. 6) drops gradually from 0.3ppbv to 0.1ppbv in the afternoon, a systematic difference evolves, with measured OH concentrations being approximately $1 \times 10^6 cm^{-3}$ higher than the model calculations. The discrepancy is of similar magnitude as the averaged unexplained OH determined in the chemical modulation experiments (Table 2). Thus, the overall agreement for OH would improve, if the unaccounted signal was fully considered as an OH measurement interference. However, the underestimation of OH would persist for low NO conditions if a potential unaccounted signal was subtracted. When NO concentrations are less than 100pptv, the observed-to-modelled OH ratio would be reduced from 1.9 to 1.5, indicating that an OH source would still be missing for low NO conditions."

**3)** **Comments:** *In their measurements of HO$_2$, the authors varied the added NO to determine the interference from alkene and aromatic peroxy radicals. However, it is unclear to me how the authors determined the RO$_2$ conversion efficiencies described on page 11 unless the absolute conversion efficiency for one of the NO flows was determined through calibrations with known concentrations of peroxy radicals. Did the authors perform RO$_2$ conversion efficiency calibrations similar to that described in Fuchs et al., 2011? This should be clarified.*

**Answers:** The NO concentration was in all cases lowered to values, at which no significant interferences from RO$_2$ in the HO$_2$ detection are expected (p10 l296-299). The NO concentration was varied between two values, in order to check, if this was

the case, because a systematic difference between measurements at the two NO concentrations is expected, if $RO_2$ was detected together with $HO_2$. For the majority of measurements, the difference was only a few percent, which means that essentially no $RO_2$ interferences were present. Nevertheless, a small correction was applied, for which was assumed that the interference from $RO_2$ was the same at all times. The correction factor was derived from a regression analysis between measurements at different NO concentrations.

**Change:** We revised the text from Line 307 to Line 311: "The $HO_2^*$ ratios were used to derive correction factors for the determination of interference-free $HO_2$ concentrations. For small NO concentrations as used in this work, we assume that the interference from $RO_2^{\#}$ is directly proportional to the applied NO concentration. Based on this assumption, we derived $HO_2^*/HO_2$ ratios of 1.02, 1.05, and 1.2 for the addition of 2.5, 5, and 20ppmv NO, respectively. These ratios were then used as correction factors to generate a consistent data set of interference-free $HO_2$ concentrations from the $HO_2^*$ measurements. After all, the correction was small enough that deviations from this assumption would not significantly affect our results."

**4)  Comments:** *It is not clear how the authors derive the $RO_2^{\#}$ concentrations and compare it to the model. The measured $HO_2^*$ in the ROx channel reflects the conversion of alkene, aromatic, and other $RO_2$ radicals to $HO_2$ in the detection cell with a conversion efficiency dependent on the $RO_2$ radical as described in Fuchs et al. (2011). Subtracting the $HO_2$ measured in the $HO_2$ axis gives $RO_2^{\#}$. Ideally, the authors should compare this measured value which is the result of various conversion efficiencies to the modeled $RO_2^{\#}$, where the individual modeled $RO_2$ concentrations are scaled by their expected conversion efficiencies, which are not necessarily all 0.8. However, it appears that the authors are scaling the measured $RO_2^{\#}$ by an average conversion efficiency of 0.8 and comparing this value to the modeled concentration of the sum of the interfering $RO_2$ concentrations. This should be clarified. Have the authors measured the individual $RO_2$ conversion efficiencies for their instrument?*

**Answers:** In the current setup, the $RO_2^{\#}$ concentrations is derived from the $HO_2$ concentrations detected in the $HO_2$ cell and the $HO_2^*$ concentrations detected in the $RO_2$ cell applying the expression $RO_2^{\#} = (HO_2^* - HO_2)/0.8$. An averaged relative

detection sensitivity for the $RO_2^{\#}$ in the $RO_2$ cell is estimated to be 0.8 according to previous publications (Fuchs et al., 2011; Lu et al., 2012) and dominant VOCs reactivity groups (mainly alkenes and isoprene) in Wangdu. We have performed tests on the $RO_2$ conversion efficiency for simple alkene, e.g. ethene, propene, for this new instrument and observed similar conversion efficiencies as Fuchs et al. (2011) reported for low NO concentrations in the detection cell. The other $RO_2$ conversion efficiencies are then extrapolated as done in Lu et al. (2012).

**Change:** We modified the text in Line 331 as ''The concentration measurements of $HO_2$ (from the $HO_2$ cell) and of $HO_2^{*}$ (from the ROx system) allow to estimate the total concentration of $RO_2^{\#}$ (Whalley et al., 2013): $RO_2^{\#} = ([HO_2^*] - [HO_2])/\alpha RO_2^{\#}$''

**5) Comments:** *The authors state that the underestimation of the $RO_2$ concentrations by the model during the high NO conditions in the morning is improved when the OH reactivity of the model is increased, but few details are provided. Similar results were found during CalNex by Griffith et al. (JGR, 2016). How much did the modeled $RO_2$ increase in this scenario? Perhaps the results of this model run could be added to Figures 5 and 9.*

**Answers:** We have performed sensitivity test to upscale the VOC reactivity, which is shown in the revised Figure 5 as following (S1). In this sensitivity test, the input VOC concentrations are scaled up so that the modelled OH reactivity agree to measurements. The observed-to-modelled $RO_2$ ratio is improved from 2.8 to 1.7 on average from 06:00 to 09:00. More detail information is given in the revised manuscript as following.

[Figure]

Figure 5    Comparison of hourly median diurnal profiles of OH, $HO_2$, $RO_2$, $RO_2^{\#}$ concentrations and $k_{OH}$ and the ozone production rate $P(O_3)$ (thick lines give median values, colored areas give 25% and 75% percentiles). S0 (blue line) denotes results from the base model run. S1 (cyan, dashed line) shows results, when the VOC concentrations in the model are increased to match the observed OH reactivity. S2 (violet, dashed line) shows results, when an additional primary $RO_2$ source ($2ppbvh^{-1}$) is added in the model for the time between 6:00 and 12:00. Grey areas indicate nighttime.

**Change:** We revised Page 18 from Line 569 to Line 596: "The strong underprediction of the observed $RO_2$ by more than a factor of 4 in the morning cannot be explained by the measurement errors and interferences discussed in Sections 2.3.4 and 2.3.5. In order to explore potential reasons for this underprediction, several sensitivity tests were performed. First, the impact of a faster OH to $RO_2$ conversion by an increased amount of VOC was tested (model sensitivity run S1). Second, an additional primary source of $RO_2$ was introduced into the chemical mechanism (S2). Third, the possibility of a slower removal rate of $RO_2$ was tested (S3).

The first possibility (S1) is supported by the observation that the modelled OH reactivity in the base run (S0) is smaller than the measured OH reactivity in the morning

until about 09:00. If this missing reactivity is caused by unmeasured VOCs, the true $RO_2$ production from reactions of VOCs with OH would be larger than the modelled one. To fill this gap, the total concentration of the measured VOCs was increased to match the measured kOH in the time window from 06:00 to 09:00. The relative partitioning of the VOCs was not changed. The model run (S1) with the upscaled VOC reactivity resolves part of the $RO_2$ discrepancy until 09:00 (Fig. 5). The observed-to-modelled $RO_2$ ratio is improved from 2.8 to 1.7 without affecting the good model-measurement agreement for OH and $HO_2$. Further sensitivity tests showed that the modelled $RO_2$ is not sensitive to the speciation of the additional VOC reactivity, since the required change of $k_{OH}$ is relatively small (< 20%). Because no missing OH reactivity was found after 09:00 h in the morning, the gap between measured and observed $RO_2$ cannot be explained by unmeasured VOCs later in the morning.

…"

**6)    Comments:** *Similarly, the authors find that the model underestimates the rate of ozone production under high NO conditions due to the underestimation of $RO_2$ radicals by the model. Similar results were found during CalNex (Brune et al., Faraday Discuss., 2016, 189, 169; Griffith et al., JGR, 2016). Does the underestimation of $RO_2$ (and therefore $P_{O3}$) depend on the measured OH reactivity? Griffith et al. (2016) found that the underestimation of $P_{O3}$ by the model was higher when the OH reactivity from VOCs was the greatest.*

**Answers:** In the sensitivity run, in which we scaled VOCs to match measured OH reactivity, also the discrepancy between modelled and calculated ozone production rate is reduced (see revised figure 5).

We tested, if there is a correlation between the underestimation of $RO_2$ by the model and VOC reactivity. However, no clear relation is observed for this campaign.

**Change:** We added discussion on ozone production underprediction found in other field campaigns, including CalNex, to show the common feature of model inability to reproduce peroxy radical concentration for high NOx condition. The text is added to the end of Section 3.8: "Other HOx field studies have also found that models underpredict the observed ozone production rate in urban atmospheres (Martinez et al., 2003; Ren et al., 2003; Kanaya et al., 2008; Mao et al., 2010; Kanaya et al., 2012; Ren

et al., 2013; Brune et al., 2016; Griffith et al., 2016). In these studies, the observed production rates were determined from measured $HO_2$ concentrations only, without the contribution of $RO_2$ for which measurements were not available. In general, the ozone production from $HO_2$ was underpredicted by chemical models at NO mixing ratios greater than 1 ppbv, reaching a factor of about 10 between 10 ppbv and 100 ppbv NO. In campaigns before 2011, unrecognized interferences from $RO_2$# species may have contributed to the deviation between measurement and model results. The interference, however, is expected to account for less than a factor of 2, because $HO_2$ and $RO_2$ concentrations are approximately equal (Cantrell et al., 2003; Mihelcic et al., 2003) and $RO_2^{\#}$ is only a fraction of the total $RO_2$ (e.g., Fig. 5). This expectation has been confirmed in recent studies, where the interference was taken into account and the significant underprediction of the ozone production from $HO_2$ still persists (Ren et al., 2013; Brune et al., 2016; Griffith et al., 2016). During the CalNex-LA 2010 campaign in Pasadena (California), part of the discrepancy could be explained by unmeasured VOCs, which were recognized as missing OH reactivity (Griffith et al., 2016). Another major reason for the $HO_2$ underprediction could be an incomplete understanding of the $HO_2$ chemistry at high NOx concentrations (Ren et al., 2013; Brune et al., 2016; Griffith et al., 2016). "

---

## Author Comment (AC3) · 25 Nov 2016

We would like to thank the reviewer for comments and questions which helped us to improve the manuscript. The reviewer comments are given below together with our responses and changes made to the manuscript.

Note: All the comments to change grammar and wordings suggested by the reviewer were changed accordingly. We appreciate the detail correction.

**Comments:** *Line 14. It is a bit confusing to say the $RO_2$ is in good agreement, but then to say that $RO_2$ is underestimated by a large amount in the morning. Suggest changing this text to make the points clearer.*

**Answers:** We changed the text Line 11:"…If additional OH recycling equivalent to 100 pptv NO is assumed, the model is capable of reproducing the observed OH, $HO_2$ and $RO_2$ concentrations for conditions of high VOC and low NOx concentrations. For $HO_2$, good agreement is found between modelled and observed concentrations at day and night. In case of $RO_2$, the agreement between model calculations and measurements is good in the late afternoon when NO concentrations are below 0.3ppbv. A significant model underprediction of $RO_2$ by a factor 3 to 5 is found in the morning at NO concentrations higher than 1ppbv, which can be explained by a missing $RO_2$ source of 2 ppbv $h^{-1}$."

**Comments:** *Line 45. Do the values of 0.8 and 0.4 ppbv refer to average or median values required? If so, state this.*

**Answers:** We changed the text Line 45: "An equivalent of 0.8 ppbv and 0.4 ppbv of NO was required in PRD and Beijing on average, respectively."

**Comments:** *Line 109. Note that peroxy radicals will also shift the NO to $NO_2$ ratio as the air travels down the sample line.*

**Answers:** We can in principle apply the inlet correction of the peroxy radicals to the NOx measurement, since we have ambient measurements of peroxy radicals. However, because the peroxy radicals are highly reactive, they are expected to be

easily lost in the inlet. Their contribution to the shift of NO to $NO_2$ is in general small compared to that of $O_3$. Therefore, a correction would not significantly change results.

We added a statement in Line 110: 'The effect of changes of the NO to $NO_2$ ratio by peroxy radicals is negligible due to their small concentrations and their high loss rate in the inlet line.'

**Comments:** *Table 1. All of the various techniques when multiple instruments were measuring are not given. Suggest making the list complete.*

**Answers:** We extended Table 1 to include all the techniques (e.g. for NOx, HONO, etc).

**Comments:** *Line 120. This discussion of HONO measurements is good, but given the potential uncertainties in such observations, it might be good to do a more detailed comparison, perhaps including a figure comparing all six measurements.*

**Answers:** A detailed comparison of the different HONO measurements is beyond the scope of this publication and will be the topic of a separate publication. Differences in the HONO measurements do not change results of our analysis here and are taken into account as additional uncertainty.

**Comments:** Line 130. An instrument with higher sensitivity is more sensitive. This is a common confusion. Suggest changing to "are generally better,"

**Answers:** We revised the text Line 128 – 136: "HONO measurements from the FZJ-LOPAP instrument are used as model constraint, because it showed the best detection limit and temporal coverage during the campaign. Results of model calculations only change less than 10%, if either measurements by the PKU LOPAP or NOAA CEAS are used as constraint. The other CEAS HONO instruments measured only during a few days. The GAC HONO measurement is known to be affected by interferences from ambient $NO_2$ and was therefore not used here."

**Comments:** *Line 220. It says that the correction is small compared to ambient OH, but this depends on the conditions. Near sunrise and sunset (or at night), this could be a large correction.*

**Answers:** We revised the text: 'A correction is applied that is small compared to ambient OH concentrations during daytime:'

**Comments:** *Line 225. It says that there was no interference from ozonolysis of simple alkenes, but what about larger, more complex alkenes (non-biogenic) that could be present?*

**Answers:** This statement summarizes results reported in Fuchs et al. (2016). The result was that ozonolysis reactions in general (most likely including also non-biogenic alkenes) are not causing significant interferences in this type of LIF instrument for atmospheric concentrations. GC measurements also suggest that the majority of alkenes were small alkene species (ethene, propene) during this campaign. The effort to investigate interferences in the OH detection will be certainly continued in the future.

**Comments:** *Line 257. This reviewer does not like the use of "titration" in this context. The authors can do as they choose, but suggest using "removal" or "conversion" efficiency rather than titration. Also suggest removing both commas on this line.*

**Answers:** We changed this to "removal efficiency".

**Comments:** *Line 271-272. The issue of mixing reagents into a flow containing HOx radicals at ambient pressure has been solved by others, particularly those make CIMS-based HOx measurements (e.g. Mauldin et al.).*

**Answers:** As stated in the text, the system was a first attempt to apply this technique in the field and needs technical improvement in the future.

**Comments:** *Line 277. It states that the titration unit caused a 5% difference in OH sensitivity. Was this applied to the data collected while it was present?*

**Answers:** Measurements with the titration system were not used as ambient OH measurements, but only to test, if there were interferences in the detection. Therefore, a change of the sensitivity only affects the quantification of a potential interference. As stated in the text, this calculation has a large uncertainty, so that a 5% change in sensitivity would be negligible.

**Comments:** *Line 307. It states that it is assumed that the contribution of $RO_2^{\#}$ scales with added NO. Is this justified by lab studies? How are the correction factors given in lines 309-310 applied? ($HO_2 = HO_2*/CF$ ?)*

**Answers:** The $RO_2$ conversion efficiency clearly increases with increasing NO concentration because a reaction of $RO_2$ with NO is required to form $HO_2$. This dependence was also shown in laboratory studies (Fuchs et al. 2011.).

We have performed tests on the $RO_2$ conversion efficiency for simple alkene, e.g. ethene, propene, for this new instrument and observed similar conversion efficiencies as Fuchs et al. (2011) reported for low NO concentrations in the detection cell. The conversion efficiency was about 10%. The other $RO_2$ conversion efficiencies are then extrapolated as done in Lu et al. (2012).

We revised the text from Line 296 to Line 299 on page 10: "A significant reduction of the relative interference from $RO_2$ can be achieved by using a smaller amount of added NO. Although less NO will cause a smaller $HO_2$ conversion efficiency, possible interferences from $RO_2$ will be even more strongly reduced because $RO_2$ conversion to OH requires one more reaction step with NO. For this reason, the NO concentration used for the conversion of $HO_2$ during this campaign was chosen to be significantly smaller ($\leq 20$ ppmv) than in previous field campaigns (500ppmv) (Lu et al.,2012, 2013). At this low concentration, it is expected that interferences from $RO_2$ become almost negligible (Fuchs et al., 2011)."

We revised the text from Line 307 to 311 on Page 11: "The $HO_2^*$ ratios were used to derive correction factors for the determination of interference-free $HO_2$

concentrations. For small NO concentrations as used in this work, we assume that the interference from $RO_2^{\#}$ is directly proportional to the applied NO concentration. Based on this assumption, we derived $HO_2^{*}/HO_2$ ratios of 1.02, 1.05, and 1.2 for the addition of 2.5, 5, and 20ppmv NO, respectively. These ratios were then used as correction factors to generate a consistent data set of interference-free $HO_2$ concentrations from the $HO2^{*}$ measurements. After all, the correction was small enough that deviations from this assumption would not significantly affect our results."

**Comments:** *Lines 334-337. Are the ambient data corrected for the artifacts as described?*

**Answers:** The background signals from the NO addition are subtracted from the $HO_2$ and $RO_2$ measurements. Artifacts caused by $NO_3$ are not subtracted from ROx measurements since there was no measurement available. The model gives an average concentration of about 10 pptv which only would case an interference that would be equivalent to $1\times10^{7}cm^{-3}$ $RO_2$ which is similar to the detection limit of the ROx measurement.

We added a statement in Line 337: "Measurements were corrected for the NO background signal, but no correction was applied for potential interferences from $NO_3$, because no $NO_3$ measurement was available. However, model calculations (see below) suggest that there was no significant interference from $NO_3$ for conditions of this campaign."

**Comments:** *Line 358. Based on the discussion, the detection limit at noon is about 1.5 x $10^6$. Is this a systematic or random effect of the solar light leakage? Can the data be corrected for this? Is this included in the overall measurement uncertainties?*

**Answers:** Sunlight is entering the measurement cell through the orifice through which ambient air is sampled into the fluorescence cell. The signal is subtracted from the total photon count rate as described Line 206-210. The detection limit is higher in the presence of sunlight because of the higher total count rate, which is only partly due to

OH fluorescence in this case. Therefore, the statistical noise (shot noise) of the OH measurement is increased.

Line 363. Do the authors believe that RACM 2 is a suitable mechanism to study detailed HOx radical chemistry? Why not use the explicit MCM mechanism, modified as you did to included updated isoprene chemistry? Perhaps add some discussion as to why the RACM 2 mechanism was selected.

**Answers:** We applied MCM and RACM in previous, similar studies and found no difference of model results for radicals (Lu et al. 2012). The likely reason for this is that the RACM mechanism is designed for ozone prediction, which is connected to the radical recycling mechanism. An explicit mechanism that includes all VOC intermediates is not required in this case. For the same reason we modified the isoprene mechanism in RACM since this impacts OH recycling.

We added in Line 368: "Previous model studies of radical chemistry showed that predictions of radical concentrations by the RACM are similar to results by explicit mechanisms like the Master Chemical Mechanism (Lu et al. 2012)."

**Comments:** *Line 389. When saying that the OH reactivity can "be well explained", suggest adding a quantitative value to the degree of agreement (within 22% or whatever).*

**Answers:** The sentence was changed to "Slightly more than 60% of the OH reactivity can be explained by the measured concentrations of CO, NOx and hydrocarbons during daytime. More than 90% of the OH reactivity can be explained, if also measured oxygenated VOC species are included (Fuchs et al., 2016b)."

**Comments:** *Line 398. What species are being referred to as "these species"? Suggest a bit more text to make it clear.*

**Answers:** Namely, most aldehydes are running free in the model. Added a sentence: " In order to avoid unrealistic accumulation of oxygenated VOC species (mostly aldehydes), …".

**Comments:** One question: why didn't the peroxy radical concentrations also increase during this time period?

**Answers:** The peroxy radical concentrations were suppressed by higher NO concentrations on this day.

**Comments:** *Lines 485-494. It appears to me that the $NO_3$ interference is sufficient to explain some or all of the nighttime signal observed.*

**Answers:** Test with a similar design LIF instrument shows that $NO_3$ could cause an OH interference. In chamber experiment, 1ppbv of $NO_3$ yielded a signal that is equivalent to an OH concentration of $1\times10^7 cm^{-3}$ (Fuchs et al. 2016). A $NO_3$ concentration of 10 pptv that is suggested by model calculations for conditions of this campaign would cause an interference that would be equivalent to an OH concentration of $1\times10^5 cm^{-3}$, which is similar to the detection limit.

The statement 'Using $NO_3$ concentrations from the model (average. 10 pptv), the expected interference would be less than $1\times10^6 cm^{-3}$ for this campaign. ' was changed to: '.., the expected interference would be $1\times10^5 cm^{-3}$ for this campaign, 5 times less than the averaged nighttime OH measurement.'

**Comments:** *Lines 495-503. Suggest a discussion and perhaps a figure showing the major contributors to the OH reactivity.*

**Answers:** The OH reactivity contribution is presented in a separate paper by Fuchs et al. The focus for this paper is to analyze the HOx chemistry and thus the OH contribution is discussed in more detail in the accompanying paper.

**Comments:** *Lines 515-516. Suggest rewording this sentence. One suggestion would be to separate the data into two equal groups rather than have this long discussion about why the two groups are not equal in size.*

**Answers:** We have divided the data into two groups and analyzed them separately.

Though the chemical conditions were slightly different, we found similar results from model-measurement comparison of radicals for the two periods. Therefore, we combined these two periods and present campaign averaged diurnal profiles.

We simplified the sentence and tried to make it more readable: "As described in Section 3.3, chemical conditions were slightly different before and after 20 June. We found similar results of model-measurement comparisons for radicals from the two periods. Therefore, the following interpretation and discussion will focus on campaign averaged diurnal profiles. "

**Comments:** Line 522. The median measure-model difference discussed is of the order of the various artifacts and interferences. Have the data been corrected for all of them before doing this comparison? If so, suggest stating this somewhere.

**Answers:** OH data is corrected for the well-known and characterized ozone interferences and no significant interference from $NO_3$ is expected as described in section 2.3.1.

The interference tests described in 3.1 were only occasionally performed and gave only an upper limit for potential additional interferences that would not change the results of our analysis of daytime OH. No correction of data is justified from these tests. Because reviewer #2 raised the same question, please refer also to the answer there.

**Comments:** *Line 523. It is not clear what is meant by "At the same time". Suggest rewording to make this clearer.*

**Answers:** The sentence is changed to "The median diurnal profiles of the measured and modelled OH concentrations agree within their errors of 10% (1σ) and 40%, respectively, from sunrise to midafternoon. When the median NO mixing ratio (cf. Fig. 6) drops gradually from 0.3ppbv to 0.1ppbv in the afternoon, a systematic difference evolves, with measured OH concentrations being approximately $1 \times 10^6 cm^{-3}$ higher than the model calculations. The discrepancy is of similar magnitude as the averaged unexplained OH determined in the chemical modulation experiments (Table 2)."

**Comments:** *Line 563. Is there any evidence of organic nitrites contributing to the enhancement of peroxy radicals in the morning? Photolysis of such species, if they exist, could contribute to the difference seen.*

**Answers:** We have no measurements of organic nitrites during this campaign. We tested including an artificial external source of $RO_2$, which could be originating from photolytic reactions. To reproduce the observed $RO_2$, 2 ppb/h of additional $RO_2$ production is required.

**Comments:** *Line 592. It states that the production rate of $RO_2$ could be underestimated, but one should also consider that the loss rate of $RO_2$ could be overestimated somehow.*

**Answers:** We also analyzed the destruction of $RO_2$ in the morning, which is dominated by the reaction with NO. The overestimation of the $RO_2$ destruction rate could be due to 1) systematic lower NO measurements; 2) segregation between NO and $RO_2$; 3) an error of lumped reaction rate constants. A sensitivity run testing the effect of this uncertainty shows that the modelled and measured $RO_2$ would agree if the reaction rate constant of $RO_2$+NO was smaller by a factor of 4. Such large change cannot be easily explained.

**Comments:** *Line 594. It states that VOC concentrations are scaled to match measurements. Which measurements? Are they the VOCs or kOH?*

**Answers:** The VOC concentrations are scaled to match measured OH reactivity. We revised the text: "To fill this gap, the total concentration of the measured VOCs was increased to match the measured $k_{OH}$ in the time window from 06:00 to 09:00. The relative partitioning of the VOCs was not changed. The model run (S1) with the upscaled VOC reactivity resolves part of the $RO_2$ discrepancy until 09:00 (Fig. 5)."

**Comments:** *Line 600. Suggest "...concentrations that are used as constraints." A thought on nighttime chemistry: if there is $NO_3$ present, then the NO concentration*

*should be very small unless the NO₃ production rate is very large. This is because of the rapid reaction between NO and NO₃. This could help with the modeling of nighttime chemistry.*

**Answers:** We had no $NO_3$ measurement in this campaign. The observed NO was usually below detection limit of the instrument (60pptv) during nighttime. In this case, the modelled $RO_2$ is high and highly variable. We tested another model scenario that forces the NO to be higher than 60pptv to limit accumulation of $RO_2$, which reduces the observed-to-modelled ratio 1.2 during the night.

**Comments:** Lines 758-769. Suggest including Ye et al in the discussion of HONO budgets.

**Answers:** We added a sentence in Line 769. "… and photolysis of particulate nitrate is proposed to be of potential importance for the tropospheric HONO production (Ye et al., 2016). "

**Comments:** *Line 770. Suggest giving an example of a reaction of OH with VOCs that do not lead to peroxy radicals.*

**Answers:** We modified the text in line 770: "Further radical terminating OH losses include reactions with unsaturated dicarbonyls (DCB1, DCB2, DCB3) and acetyl nitrate species (PAN, MPAN, etc) in RACM2."

**Comments:** *Line 832. Yes, the interference would be minor compared to the daytime maximum, but it could be very important at sunrise and sunset.*

**Answers:** The statement was modified accordingly.

**Comments:** References. The papers on HOx measurements are very Euro-centric. Suggest adding some papers from US HOx measurement groups.

**Answers:** More results from the HOx groups outside Europe were added such as Griffith et al., 2013, 2016; Mauldin et al., 1999; Kim et al., 2014; Brune et al., 2016,

Kanaya et al., 2008, 2012.

**Comments:** *Figure 1. The colors for SN2 and SOH are very similar. Suggest changing one of them to a very different color.*

**Answers:** Changed accordingly.

**Comments:** *Figures 3, 4, 5, 6, 8, and 10. Is the gray period meant to signify nighttime? If so, the authors should check this carefully. It appears that there are photolysis processes (such as $O_3$) that occur after sunset (see Figure 10).*

**Answers:** The gray area indicates nighttime. For Figure 10 there is a typo error in the data analysis routine, we have now revised this.